# Moirai-MoE: Empowering Time Series Foundation Models with Sparse Mixture of Experts

Xu Liu [1 2]   Juncheng Liu [1]   Gerald Woo [1]   Taha Aksu [1]   Yuxuan Liang [3]   Roger Zimmermann [2]   Chenghao Liu [1]
Junnan Li [1]   Silvio Savarese [1]   Caiming Xiong [1]   Doyen Sahoo [1]

## Abstract

Achieving effective unified pretraining on large time series corpora remains an open challenge in developing time series foundation models. Existing methods, such as MOIRAI, introduce multiple projection layers for time series of different frequencies to account for high data heterogeneity. We identify major drawbacks to this human-imposed frequency-level model specialization. First, frequency is not a reliable indicator for grouping pretraining data. Second, time series can display varied distributions even within a short window. Frequency-level specialization overlooks the diversity at this granularity. To address these issues, this paper introduces MOIRAI-MOE, excluding human-defined data groupings while delegating the modeling of diverse time series patterns to the sparse mixture of experts (MoE) within Transformers. With this design, MOIRAI-MOE eliminates reliance on heuristics and enables automatic token-level specialization. Extensive evaluations on 39 datasets demonstrate the superiority of MOIRAI-MOE over state-of-the-art foundation models. This study also conducts comprehensive model analyses to explore the inner workings of time series MoE foundation models.

## 1. Introduction

Time series forecasting is experiencing a major shift (Liang et al., 2024). The traditional approach of developing separate models for each dataset is being replaced by the concept of universal forecasting (Woo et al., 2024), where a pretrained foundation model can be applied across diverse downstream forecasting tasks in a zero-shot manner, regardless of variations in domain, frequency, dimensionality, context, or prediction length. This new paradigm significantly reduces the complexity of building numerous specialized models, paving the way for forecasting-as-a-service.

However, unlike language and vision modalities which benefit from standardized input formats, time series corpora are highly heterogeneous, posing significant challenges during model pretraining. Existing solutions such as UniTime (Liu et al., 2024a) and TEMPO (Cao et al., 2024) leverage language prompts to discern the source of data, thereby achieving dataset-level model specialization. MOIRAI (Woo et al., 2024) goes a step further and proposes a more granular categorization based on a time series meta feature – frequency. Specifically, they design multiple input/output projection layers with each layer specialized to handle data from a specific frequency, thereby enabling frequency-level specialization. TimesFM (Das et al., 2024) is at this level as well, grouping data via a frequency embedding mapping.

Given the heterogeneity of time series, we acknowledge the value of model specialization; however, we argue that *human-imposed frequency-level specialization lacks generalizability and introduces several limitations*. First, frequency is not a reliable indicator for grouping pretraining corpora. As shown in Figure 1, time series with different frequencies can exhibit similar patterns, while those with the same frequency may display diverse and unrelated patterns. This human-imposed mismatch between frequency and pattern undermines the efficacy of model specialization. Furthermore, real-world time series are inherently non-stationary (Liu et al., 2022), displaying varied distributions even within a short window. Frequency-level specialization overlooks the diversity at this granularity, underscoring the need for more fine-grained modeling approaches.

To tackle the aforementioned issues, this paper introduces **MOIRAI-MOE**, an innovative solution for *effective and efficient time series pretraining*, inspired by recent developments of Sparse Mixture of Experts (MoE) Transformers (Lepikhin et al., 2021; Fedus et al., 2022; Dai et al., 2024). The core idea of MOIRAI-MOE is to exclude human-defined time series groupings while delegating the model-

---

[1]Salesforce AI Research [2]National University of Singapore [3]The Hong Kong University of Science and Technology (Guangzhou). Correspondence to: Chenghao Liu <chenghao.liu@salesforce.com>.

*Proceedings of the $42^{nd}$ International Conference on Machine Learning*, Vancouver, Canada. PMLR 267, 2025. Copyright 2025 by the author(s).

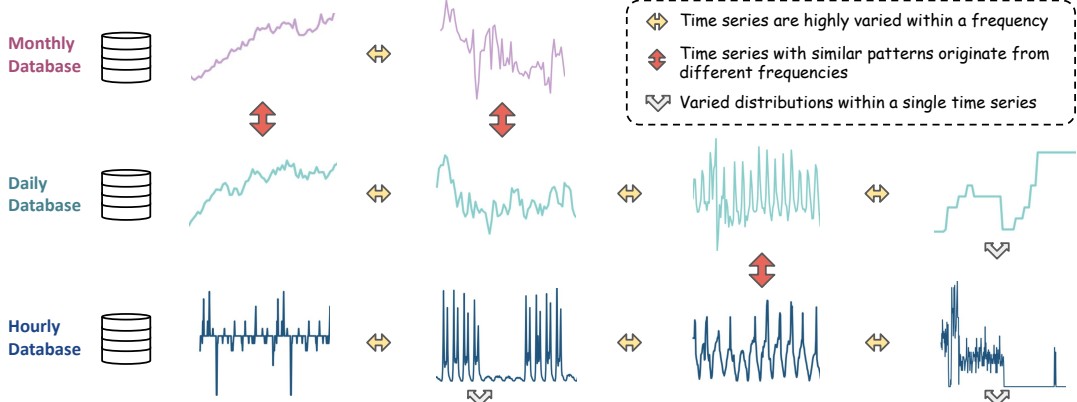

*Figure 1.* An illustration of the challenges arising from grouping time series by frequency and imposing frequency-level model specialization: the diversity of patterns within the same frequency group, the similarity of patterns across different frequencies, and the variability of distributions within a single time series. The examples presented are **real time series** in the Monash benchmark (Godahewa et al., 2021).

ing of diverse time series patterns to the sparsely activated experts in Transformer layers. With this design, specialization of MOIRAI-MOE is achieved in a data-driven manner and operates at the time series token level. It also benefits from the enhanced computational efficiency and scalability brought by the MoE architecture. Moreover, this study investigates existing expert gating functions that generally use a randomly initialized linear layer for expert assignments (Shazeer et al., 2017; Jiang et al., 2024) and proposes a new function that leverages cluster centroids derived from an existing pretrained dense model to guide expert allocations.

We extensively evaluate MOIRAI-MOE using a total of 39 datasets in in-distribution and zero-shot forecasting scenarios. The results confirm the superiority of MOIRAI-MOE over state-of-the-art foundation models. Additionally, we conduct comprehensive model analyses, as the first attempt, to explore the inner workings of time series MoE foundation models. It reveals that MOIRAI-MOE acquires the capability to achieve frequency-invariant representations and essentially performs progressive denoising throughout the model. Our contributions are summarized as follows:

- We propose MOIRAI-MOE, a pioneering mixture-of-experts time series foundation model, achieving token-level model specialization in a data-driven manner. We introduce a new expert gating function for accurate expert assignments and improved performance.

- Extensive experiments on 39 datasets reveal that MOIRAI-MOE delivers up to 17% performance improvements over MOIRAI at the same level of model size, and outperforms other time series foundation models with up to 65× fewer activated parameters.

- We conduct thorough model analyses to deepen understanding of the inner workings of time series MoE foundation models and summarize insights for future research.

## 2. Related Work

**Module Specialization for Time Series Foundation Models**   A key challenge in pretraining time series foundation models lies in accommodating high data diversity, underscoring the need for designing specialization modules. Approaches like UniTime (Liu et al., 2024a) and TEMPO (Cao et al., 2024) utilize prompts to identify data sources, facilitating dataset-level specialization. MOIRAI (Woo et al., 2024) and TimesFM (Das et al., 2024) refine this further by grouping data based on a time series meta feature – frequency. While Moment (Goswami et al., 2024) and UniTS (Gao et al., 2024) employ different projection heads for various time series analysis tasks, including classification and anomaly detection. Methods like Chronos (Ansari et al., 2024), Lag-LLaMA (Rasul et al., 2023), and Timer (Liu et al., 2024c) lack specialization modules, potentially increasing learning complexity and requiring more parameters to handle diverse inputs. In this work, we propose automatic time series token-level specialization, where diverse tokens are handled by different experts, while similar tokens share parameter space, reducing learning complexity.

**Mixture-of-Experts Time Series Models**   In vision and language, sparse mixture of experts (MoE) has proven effective for scaling Transformer capacity while minimizing computational overheads (Fedus et al., 2022; Dai et al., 2024; Zhu et al., 2024). The application of Mixture of Experts (MoE) in time series research remains limited, with only a few studies exploring MoE concepts in dataset-specific training (Ismail et al., 2022; Ni et al., 2024). In these studies, the experts are typically linear models, such as autoregressive models and DLinear (Zeng et al., 2023), which are small-scale models. In contrast, MOIRAI-MOE represents a time series foundation model capable of scaling efficiently without a significant increase in computational costs. A concurrent work, Time-MoE (Shi et al., 2024), also integrates MoE

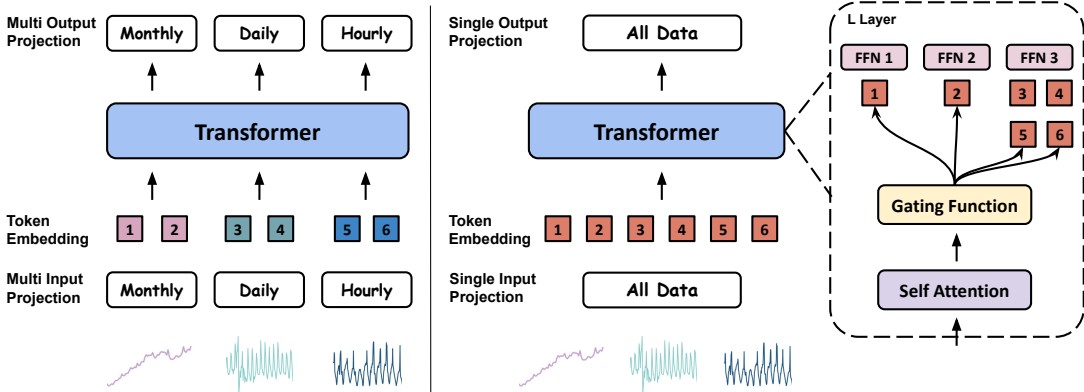

*Figure 2.* Comparison of MOIRAI (left) and MOIRAI-MOE (right).

into time series foundation models. However, it relies on existing gating functions without proposing new alternatives or investigating the internal mechanisms of MoE models. Additionally, Time-MoE employs point-wise tokenization, which may fail to capture the underlying semantics of time series data (Zeng et al., 2023), limiting its performance.

## 3. Methodology

In this section, we present MOIRAI-MOE, a scalable time series foundation model built upon MOIRAI (Woo et al., 2024). Figure 2 presents a comparison. MOIRAI-MOE significantly improves MOIRAI by removing the heuristically defined input/output projection layers that handle data of varying frequencies. Instead, MOIRAI-MOE leverages sparse mixture of experts within Transformers to autonomously capture the diverse patterns in time series data, allowing for greater flexibility. In addition, MOIRAI-MOE proposes a novel gating function that routes tokens based on the knowledge of an existing pretrained dense model, and adopts the next-token prediction objective to improve pretraining efficiency by enabling parallel learning of various context lengths in a single model update.

### 3.1. Time Series Token Construction

Patching techniques, first introduced in PatchTST (Nie et al., 2023), have become a prevalent method in time series forecasting (Das et al., 2024; Liu et al., 2024a; Woo et al., 2024). By aggregating adjacent time series data into patches, this technique effectively captures local semantic information and significantly reduces computational overhead when processing long inputs. Given a time series with length $S$, we segment it into non-overlapping patches of size $P$, resulting in a sequence of patches $\boldsymbol{x} \in \mathbb{R}^{N \times P}$, where $N = \lceil \frac{S}{P} \rceil$.

We then normalize the patches to mitigate distribution shift issues (Liu et al., 2022; Wu et al., 2023). In a decoder-only (autoregressive) model, where each patch predicts its suc-

ceeding patch, applying a causal normalizer to each patch is the most effective way to achieve accurate normalization. However, this approach generates $N$ subsequences with different lengths, diminishing the parallel training that decoder-only models typically offer. To address this, we introduce the masking ratio $r$ as a hyperparameter, which specifies the portion of the entire sequence used exclusively for robust normalizer calculation, without contributing to the prediction loss. Finally, we forward the patches through a single projection layer to generate time series tokens $\boldsymbol{x} \in \mathbb{R}^{N \times D}$, where $D$ is the dimension of Transformers. We pass on the capability of learning diverse time series patterns to the vast number of parameters in Transformers. This projection layer is implemented as a residual multi-layer perceptron to enhance representation capacity (Das et al., 2023).

### 3.2. Sparse Mixture-of-Experts Transformers

A decoder-only Transformer (Dubey et al., 2024) is constructed by stacking $L$ layers of Transformer blocks. The block at the $l$-th layer is represented as follows:

$$\tilde{\boldsymbol{x}}^l = \text{CSA}(\text{LN}(\boldsymbol{x}^l)) + \boldsymbol{x}^l \tag{1}$$

$$\boldsymbol{x}^{l+1} = \text{FFN}(\text{LN}(\tilde{\boldsymbol{x}}^l)) + \tilde{\boldsymbol{x}}^l \tag{2}$$

where CSA, FFN, and LN denote a causal self-attention module, a feed-forward network, and the layer normalization, respectively. $\tilde{\boldsymbol{x}}^l \in \mathbb{R}^{N \times D}$ are the hidden states of tokens after the attention module of the $l$-th layer and $\boldsymbol{x}^l = \boldsymbol{x}^{l+1} \in \mathbb{R}^{N \times D}$ are the input and output hidden states of the $l$-th layer. Note that MOIRAI-MOE considers multivariate correlations by flattening all variates into a sequence. During causal attention, each token attends to both its preceding tokens and those from other variates, enabling the model to capture intra- and inter-variate dependencies.

Next, we establish the mixture of experts by replacing each FFN with a MoE layer, which is composed of $M$ expert networks $\{E_1, \ldots, E_M\}$ and a gating function $G$. Only a subset of experts is activated for each token, allowing

experts to specialize in distinct patterns of time series data and ensuring computational efficiency. The output of the MoE layer is computed as:

$$\sum_{i=1}^{M} G(\tilde{\boldsymbol{x}}^l)_i \cdot E_i(\tilde{\boldsymbol{x}}^l) \tag{3}$$

where $E_i(\tilde{\boldsymbol{x}}^l)$ is the output of the $i$-th expert network, and $G(\tilde{\boldsymbol{x}}^l)_i$ is the $i$-th token-to-expert affinity score generated by the gating function. Following (Lepikhin et al., 2021; Zhu et al., 2024; Jiang et al., 2024), we set the number of activated experts to $K = 2$.

### 3.2.1. GATING FUNCTION

The most common linear gating mechanism projects MoE inputs $\tilde{\boldsymbol{x}}^l$ by weights $\boldsymbol{W}_g \in \mathbb{R}^{D \times M}$, then selects the top-K logits and applies the softmax function over them (Shazeer et al., 2017; Jiang et al., 2024; Dai et al., 2024):

$$G(\tilde{\boldsymbol{x}}^l) = \text{Softmax}(\text{TopK}(\tilde{\boldsymbol{x}}^l \cdot \boldsymbol{W}_g)) \tag{4}$$

To mitigate the load balancing issue (Shazeer et al., 2017) with this sparse gating, an auxiliary loss is typically applied to encourage an even distribution of tokens across experts (Lepikhin et al., 2021; Jiang et al., 2024; Dai et al., 2024). Formally, the load balancing loss for a batch $\mathcal{B}$ containing $T$ tokens is defined as:

$$\mathcal{L}_{\text{load}} = \sum_{i=1}^{M} \mathcal{D}_i \mathcal{P}_i, \ \mathcal{D}_i = \frac{1}{T} \sum_{t=1}^{T} \mathbb{1}\{\text{Token t selects Expert i}\},$$
$$\tag{5}$$

$$\mathcal{P}_i = \frac{1}{T} \sum_{t=1}^{T} G(\tilde{\boldsymbol{x}}^l)_i \tag{6}$$

where $\mathbb{1}$ is the indicator function, $\mathcal{D}_i$ denotes the fraction of tokens routed to expert $i$, and $\mathcal{P}_i$ indicates the proportion of the gating probability allocated to expert $i$. The loss $\mathcal{L}_{\text{load}}$ is applied to each Transformer layer $l$. The loss is then aggregated by computing the mean across all layers and added to the prediction loss $\mathcal{L}_{\text{pred}}$ with a weight of 0.01 (Jiang et al., 2024; Dai et al., 2024).

While effective, these gating functions are learned entirely from scratch, and poor initialization may negatively impact their performance. Recent advances in developing MoE leveraged pretrained dense models to introduce inductive bias for enhancing performance (Komatsuzaki et al., 2022; Qwen, 2024). Inspired by these successes, we propose a novel gating mechanism using cluster centroids derived from a dense model to guide expert allocations. The intuition is that clusters of pretrained token embeddings more closely reflect the real distribution of data, leading to more effective expert specialization and improved performance.

**Token Clusters as Gating Function** Technically, we first pretrain a MOIRAI model using single-patch input/output projection layers to remove the human-imposed frequency biases in MOIRAI. We then perform inference using our pretraining data LOTSA (Woo et al., 2024). For a batch $\mathcal{B}$ containing $T$ tokens, we extract the attention outputs $\tilde{\boldsymbol{x}}^l \in \mathbb{R}^{T \times D}$ at each layer and perform mini-batch k-means clustering on them to continuously update cluster centroids $\boldsymbol{C}_l \in \mathbb{R}^{M \times D}$ at each layer, where the number of clusters is set to match the total number of experts. During MoE pretraining, for each layer, each token computes the Euclidean distance to the cluster centroids, and these distances serve as token-to-expert affinity scores for expert assignments:

$$G(\tilde{\boldsymbol{x}}^l) = \text{Softmax}(\text{TopK}(\text{Euclidean}(\tilde{\boldsymbol{x}}^l, \boldsymbol{C}_l))) \tag{7}$$

### 3.3. Pretraining Objective

Let $\boldsymbol{x}_{t-l+1:t} = \{\boldsymbol{x}_{t-l+1}, \ldots, \boldsymbol{x}_t\}$ denote the context window of length $l$ for a token at position $t$. In this study, to facilitate both point and probabilistic forecasting, our goal is formulated as forecasting the distribution of the next token $p(\boldsymbol{x}_{t+1}|\phi)$ by predicting the mixture distribution parameters $\hat{\phi}$ (Woo et al., 2024). These parameters are derived from the output tokens of the Transformer, followed by a single output projection layer. The following negative log-likelihood is minimized during pretraining:

$$\mathcal{L}_{\text{pred}} = -\log p(\boldsymbol{x}_{t+1}|\hat{\phi}), \ \hat{\phi} = f_{\boldsymbol{\theta}}(\boldsymbol{x}_{t-l+1:t}) \tag{8}$$

## 4. Experiments

### 4.1. MOIRAI-MOE Setup

To ensure a fair comparison with MOIRAI in terms of activated parameters, we set the number of activated experts to 2 for MOIRAI-MOE, resulting in 11M/86M activated parameters per token for MOIRAI-MOE$_S$/MOIRAI-MOE$_B$, closely matching the dense model MOIRAI$_S$/MOIRAI$_B$ that contains 14M/91M activated parameters. The total number of experts is set to 32, yielding total parameter sizes of 117M for MOIRAI-MOE$_S$ and 935M for MOIRAI-MOE$_B$. MOIRAI-MOE$_L$ is not presented due to the significant requirements of computational resources. The specific configurations are outlined in Table 1. See the effects of the number of MoE layers in Appendix B.2.

Table 1. Model configurations of MOIRAI and MOIRAI-MOE.

| Model | Layers | $d_{\text{model}}$ | $d_{\text{ff}}$ | Activated Params | Total Params |
|---|---|---|---|---|---|
| MOIRAI$_S$ | 6 | 384 | 1,024 | 14M | 14M |
| MOIRAI$_B$ | 12 | 768 | 2,048 | 91M | 91M |
| MOIRAI$_L$ | 24 | 1,024 | 2,736 | 310M | 310M |
| MOIRAI-MOE$_S$ | 6 | 384 | 512 | 11M | 117M |
| MOIRAI-MOE$_B$ | 12 | 768 | 1,024 | 86M | 935M |

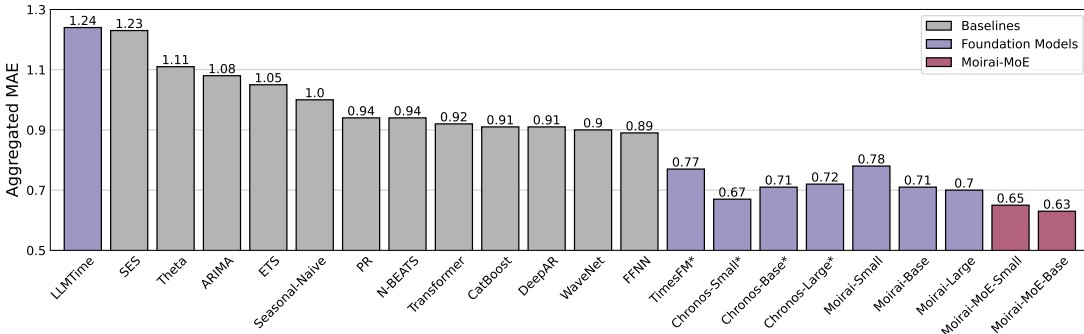

*Figure 3.* In-distribution evaluation using 29 datasets from the Monash benchmark (Godahewa et al., 2021). * indicates that the methods includes the test datasets here in their pretraining corpora. The aggregated MAE is reported, with each dataset's MAE normalized by seasonal naive, followed by geometric mean. Dataset details, full results, and context lengths are in Appendix A.1 and A.4.

## 4.2. Main Results

**In-distribution Forecasting** We begin with an in-distribution evaluation using a total of 29 datasets from the Monash benchmark (Godahewa et al., 2021). Their training set are included in LOTSA (Woo et al., 2024), holding out the test set which we now use for assessments. Figure 3 summarizes the results based on the aggregated mean absolute error (MAE), in comparison with the baselines presented in the Monash benchmark and following foundation models: LLMTime (Gruver et al., 2023), TimesFM (Das et al., 2024), Chronos (Ansari et al., 2024), and MOIRAI (Woo et al., 2024). Dataset details and full results are provided in Appendix A.1. **The evaluation results show that MOIRAI-MOE beats all competitors.** In particular, MOIRAI-MOE$_S$ drastically surpasses its dense counterpart MOIRAI$_S$ by 17%, and outperforms the larger MOIRAI$_B$ and MOIRAI$_L$ by 8% and 7%, respectively. MOIRAI-MOE$_B$ delivers a further 3% improvement over MOIRAI-MOE$_S$. Compared to Chronos, which MOIRAI could not surpass, MOIRAI-MOE successfully bridges the gap and delivers superior results with up to 65× fewer activated parameters.

**Zero-shot Forecasting** Next, we conduct an out-of-distribution evaluation on 10 datasets not included in LOTSA (see dataset details in Appendix A.2). To establish a comprehensive comparison, we report results for both probabilistic and point forecasting using 6 metrics. The continuous ranked probability score (CRPS) and mean absolute scaled error (MASE) are shown in Table 2, with additional metrics in Table 9 and 10. For baselines, we compare against foundation models: TTM (Ekambaram et al., 2024), Timer (Liu et al., 2024c), TimesFM, Chronos, MOIRAI, and Time-MoE (Shi et al., 2024), as well as state-of-the-art full-shot models trained on individual datasets: TiDE (Das et al., 2023), PatchTST (Nie et al., 2023), iTransformer (Liu et al., 2024b), and MoLE-DLinear (Ni et al., 2024). Context lengths of all methods are provided in Appendix A.4. Due to page limits, partial baselines are presented in Table 2 and full results are in Table 8. **MOIRAI-MOE$_B$ achieves the best over-**

**all zero-shot performance, outperforming baselines that included partial test data in their pretraining corpora.** When compared to all sizes of MOIRAI, MOIRAI-MOE$_S$ delivers a 3%–14% improvement in CRPS and an 8%–16% improvement in MASE. These improvements are remarkable, considering that MOIRAI-MOE$_S$ has only 11M activated parameters – 28× fewer than MOIRAI$_L$.

**Summary** Our extensive evaluation validates the effectiveness of MOIRAI-MOE's design and showcases its strong generalization abilities. It also highlights the superiority of token-level specialization over frequency-based approaches (TimesFM, MOIRAI) and models without specialization modules (Chronos, TTM). Additionally, MOIRAI-MOE significantly outperforms Time-MoE, underscoring the advantages of cluster-based gating and patch-based tokenization.

## 4.3. Ablation Studies

**Model Design** In the main results, we simultaneously enable the mixture of experts and switch the pretraining objective from a masked filling approach to a next-token prediction approach. To ensure a more rigorous comparison, we conduct further experiments where only the learning objective is changed. Table 3 presents the Monash evaluation results, with the first and last rows representing MOIRAI$_S$ and MOIRAI-MOE$_S$, respectively. This outcome suggests that altering the learning objective alone yields modest performance improvements, while the major gains stem from leveraging experts for automatic token-level specialization.

**Pretraining Objective** We adopt the next-token prediction objective for its superior pretraining efficiency compared to the masked filling. To illustrate this, we conduct experiments with varying pretraining steps, as shown in Figure 4 (left). The results show that the next-token prediction approach consistently outperforms masked filling at each evaluated step. Moreover, next-token prediction pretraining with 50k steps achieves comparable performance

*Table 2.* Zero-shot performance comparisons with * indicates the non-zero-shot datasets. The Avg column is computed by normalizing each dataset's MAE by seasonal naive, then applying the geometric mean. Two Avg values are shown: one that averages all, and another (non-leak) excludes Electricity and Solar. Best results are in red. Second best results are in blue. Power: Turkey Power. Traffic: Istanbul Traffic. Weather: Jena Weather. BizITObs: BizITObs-L2C. Dataset details, full results, and context lengths are in Appendix A.2 and A.4.

| Method | Metric | Electricity | Solar | Power | ETT1 | ETT2 | Traffic | MDENSE | Walmart | Weather | BizITObs | Avg (all) | Avg (non-leak) |
|---|---|---|---|---|---|---|---|---|---|---|---|---|---|
| Seasonal Naive | CRPS | 0.070 | 0.512 | 0.085 | 0.515 | 0.205 | 0.257 | 0.294 | 0.151 | 0.068 | 0.262 | 1.000 | 1.000 |
| | MASE | 0.881 | 1.203 | 0.906 | 1.778 | 1.390 | 1.137 | 1.669 | 1.236 | 0.782 | 0.986 | 1.000 | 1.000 |
| iTransformer | CRPS | 0.057 | 0.443 | 0.056 | 0.344 | 0.129 | 0.105 | 0.072 | 0.070 | 0.053 | 0.077 | 0.540 | 0.483 |
| | MASE | 0.875 | 1.342 | 1.076 | 2.393 | 1.841 | 0.581 | 0.727 | 0.761 | 0.623 | 0.271 | 0.767 | 0.708 |
| TTM | CRPS | 0.075 | 0.534* | 0.059 | 0.417 | 0.122 | 0.210 | 0.150 | 0.192 | 0.055 | 0.102 | 0.758 | 0.697 |
| | MASE | 0.802 | 1.255* | 0.898 | 1.934 | 1.547 | 0.901 | 1.195 | 1.477 | 0.506 | 0.308 | 0.831 | 0.798 |
| TimesFM | CRPS | 0.045* | 0.456 | 0.037 | 0.280 | 0.113 | 0.131 | 0.070 | 0.067 | 0.042 | 0.080 | **0.488** | **0.439** |
| | MASE | 0.655* | 1.391 | 0.851 | 1.700 | 1.644 | 0.678 | 0.702 | 0.735 | 0.440 | 0.310 | 0.689 | 0.640 |
| Chronos$_S$ | CRPS | 0.043* | 0.389* | 0.038 | 0.360 | 0.097 | 0.124 | 0.087 | 0.079 | 0.089 | 0.087 | 0.543 | 0.513 |
| | MASE | 0.629* | 1.193* | 0.717 | 1.799 | 1.431 | 0.622 | 0.834 | 0.849 | 0.606 | 0.301 | 0.694 | 0.661 |
| Chronos$_B$ | CRPS | 0.041* | 0.341* | 0.039 | 0.387 | 0.092 | 0.109 | 0.075 | 0.080 | 0.058 | 0.084 | 0.499 | 0.471 |
| | MASE | 0.617* | 1.002* | 0.722 | 1.898 | 1.265 | 0.553 | 0.712 | 0.849 | 0.583 | 0.301 | **0.656** | 0.631 |
| Chronos$_L$ | CRPS | 0.041* | 0.339* | 0.038 | 0.404 | 0.091 | 0.117 | 0.075 | 0.073 | 0.062 | 0.084 | 0.500 | 0.473 |
| | MASE | 0.615* | 0.987* | 0.702 | 1.959 | 1.270 | 0.597 | 0.724 | 0.788 | 0.601 | 0.310 | 0.660 | 0.638 |
| MOIRAI$_S$ | CRPS | 0.072 | 0.471 | 0.048 | 0.275 | 0.101 | 0.173 | 0.084 | 0.103 | 0.049 | 0.081 | 0.578 | 0.507 |
| | MASE | 0.981 | 1.465 | 0.948 | 1.701 | 1.417 | 0.990 | 0.836 | 1.048 | 0.521 | 0.301 | 0.798 | 0.726 |
| MOIRAI$_B$ | CRPS | 0.055 | 0.419 | 0.040 | 0.301 | 0.095 | 0.116 | 0.104 | 0.093 | 0.041 | 0.078 | 0.520 | 0.467 |
| | MASE | 0.792 | 1.292 | 0.888 | 1.736 | 1.314 | 0.644 | 1.101 | 0.964 | 0.487 | 0.291 | 0.736 | 0.685 |
| MOIRAI$_L$ | CRPS | 0.050 | 0.406 | 0.036 | 0.286 | 0.094 | 0.112 | 0.095 | 0.098 | 0.051 | 0.079 | 0.514 | 0.467 |
| | MASE | 0.751 | 1.237 | 0.870 | 1.750 | 1.436 | 0.631 | 0.957 | 1.007 | 0.515 | 0.285 | 0.729 | 0.685 |
| Time-MoE$_B$ | CRPS | 0.051* | 0.230* | 0.044 | 0.392 | 0.125 | 0.152 | 0.099 | 0.100 | 0.070 | 0.112 | 0.583 | 0.586 |
| | MASE | 0.587* | 0.535* | 0.800 | 1.823 | 1.672 | 0.672 | 0.846 | 0.833 | 0.558 | 0.343 | 0.662 | 0.695 |
| Time-MoE$_L$ | CRPS | 0.051* | 0.294* | 0.045 | 0.386 | 0.131 | 0.172 | 0.090 | 0.097 | 0.058 | 0.111 | 0.589 | 0.576 |
| | MASE | 0.581* | 0.689* | 0.790 | 1.773 | 1.878 | 0.762 | 0.759 | 0.817 | 0.524 | 0.337 | 0.678 | 0.695 |
| MOIRAI-MoE$_S$ | CRPS | 0.046 | 0.429 | 0.036 | 0.288 | 0.093 | 0.108 | 0.071 | 0.090 | 0.056 | 0.081 | 0.497 | **0.450** |
| | MASE | 0.719 | 1.222 | 0.737 | 1.750 | 1.248 | 0.563 | 0.746 | 0.927 | 0.476 | 0.298 | 0.670 | **0.620** |
| MOIRAI-MoE$_B$ | CRPS | 0.041 | 0.382 | 0.034 | 0.296 | 0.091 | 0.100 | 0.071 | 0.088 | 0.057 | 0.079 | **0.478** | **0.439** |
| | MASE | 0.638 | 1.161 | 0.725 | 1.748 | 1.247 | 0.510 | 0.721 | 0.918 | 0.509 | 0.290 | **0.651** | **0.611** |

to masked filling with 100k steps, highlighting the substantial efficiency gains provided by the next-token prediction objective. In Appendix B.1, we present further results illustrating how MOIRAI-MOE's downstream performance evolves during pretraining, compared to MOIRAI.

**Gating Function** In Figure 4 (right), we vary the total number of experts and examine the impact of gating functions on performance. Across all gating functions, performance consistently improves as the number of experts increases until 32. Moreover, our proposed token clustering method proves to be consistently superior to other gating variants across all expert configurations. This indicates that the clustering approach aligns closely with the distribution of data representations that have been optimized in pretraining, leading to more effective expert specialization than randomly initialized gating.

*Table 3.* Model variants performance on Monash.

| Model Variant | Aggregated MAE |
|---|---|
| Multi Projection w/ Masked Filling | 0.78 |
| Multi Projection w/ Next-Token Prediction | 0.75 |
| Single Projection & MoE w/ Next-Token Prediction | 0.65 |

## 4.4. Model Analyses

In this section, we delve deeper into the learned token embeddings and expert assignment distribution of MOIRAI-MOE to shed light on the inner workings of the time series MoE foundation model.

**Obs 1: MOIRAI-MOE produces token embeddings in a data-driven way, effectively improving performance.** In Figure 5, we utilize the T-SNE visualization (Van der Maaten & Hinton, 2008) to compare the token embeddings generated from the input projection layers of MOIRAI and MOIRAI-MOE. (1) In the first row, we examine the NN5 Daily and Traffic Hourly datasets, which have different frequencies but exhibit similar underlying patterns (visualizations of these patterns are in Appendix D). The figure illustrates that MOIRAI produces distinct embeddings due to the use of separate frequency projection layers, while MOIRAI-MOE successfully blends their representations together. Their inherent similarities are further demonstrated by their comparable expert distributions in the last two columns. (2) In the second row, we analyze another daily frequency dataset, Covid Daily Deaths, which shows dis-

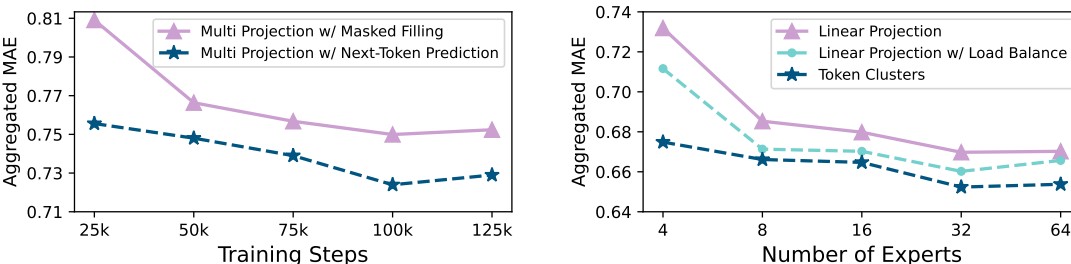

*Figure 4.* Ablation studies of the pretraining objective and gating function using MOIRAI-MOE$_S$.

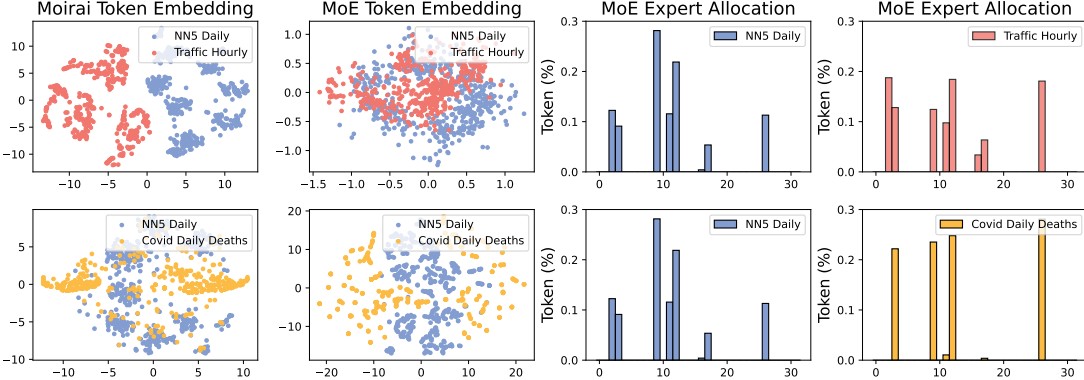

*Figure 5.* The first two columns are the comparison of embeddings from MOIRAI$_S$ and MOIRAI-MOE$_S$. The last two columns are the expert assignment distributions of MOIRAI-MOE$_S$ in layer 1: the x-axis corresponds to the 32 experts in a layer, and the y-axis is the proportion of tokens that choose experts.

tinct patterns compared to NN5 Daily. We observe that the embeddings of these two datasets overlap to some extent in MOIRAI but are effectively separated in MOIRAI-MOE. Moreover, the Covid Daily dataset shows a different expert distribution than NN5 Daily due to different token embeddings. **The data-driven modeling paradigm of MOIRAI-MOE ultimately leads to significant performance boosts**, reducing the NN5 Daily MAE from 5.37 to 4.04 (a 25% improvement), the Traffic Hourly MAE from 0.02 to 0.013 (a 35% improvement), and the Covid Daily Deaths MAE from 124.32 to 119 (a 4% improvement).

**Obs 2: Different frequency data exhibit different expert selection distributions at shallow layers but similar distributions at deep layers.** We present the expert allocation distributions on Monash grouped by frequency in Figure 6. In the shallow layers, expert selection is notably diverse across frequencies. While by the final layer (layer 6), expert allocation becomes nearly identical across all frequencies, suggesting that the model has abstracted time series into high-level representations largely independent of the frequency. This evidence indicates that **MOIRAI-MOE effectively achieves frequency-invariant hidden representations**, which are crucial for model generalization ([Van Ness et al., 2023](#)). The shared parameter space in the last layer also shows that it is sufficient for generating representations needed to make diverse predictions. Additionally, we recog-

nize that some experts in MOIRAI-MOE are rarely selected during inference on Monash. Pruning these underutilized experts for inference efficiency is left for future work.

**Obs 3: Shallow layers have more routing preferences than deep layers.** According to Figure 6, in the shallow layers, the model relies on multiple experts to manage the high level of short-term variability, such as cyclical, seasonal, or abrupt changes. As tokens are aggregated in deeper layers, the model shifts its focus to more generalizable temporal dependencies, such as broader trends and long-term patterns, that can be shared across different frequencies and leads to more concentrated experts being selected. This behavior contrasts with patterns observed in LLMs ([Zhu et al., 2024](#)), where earlier layers typically concentrate on a limited number of experts to capture common linguistic features, while deeper layers target more task-specific characteristics. This divergence may stem from the dynamic and noisier nature of time series tokens, which are generated from small time windows, unlike language tokens derived from a fixed vocabulary. **Our findings suggest that denoising processes occur progressively throughout the model**. This observation aligns with conclusions from GPT4TS ([Zhou et al., 2023](#)), which found that as the layer depth increases, token vectors are projected into the low-dimensional top eigenvector space of input patterns.

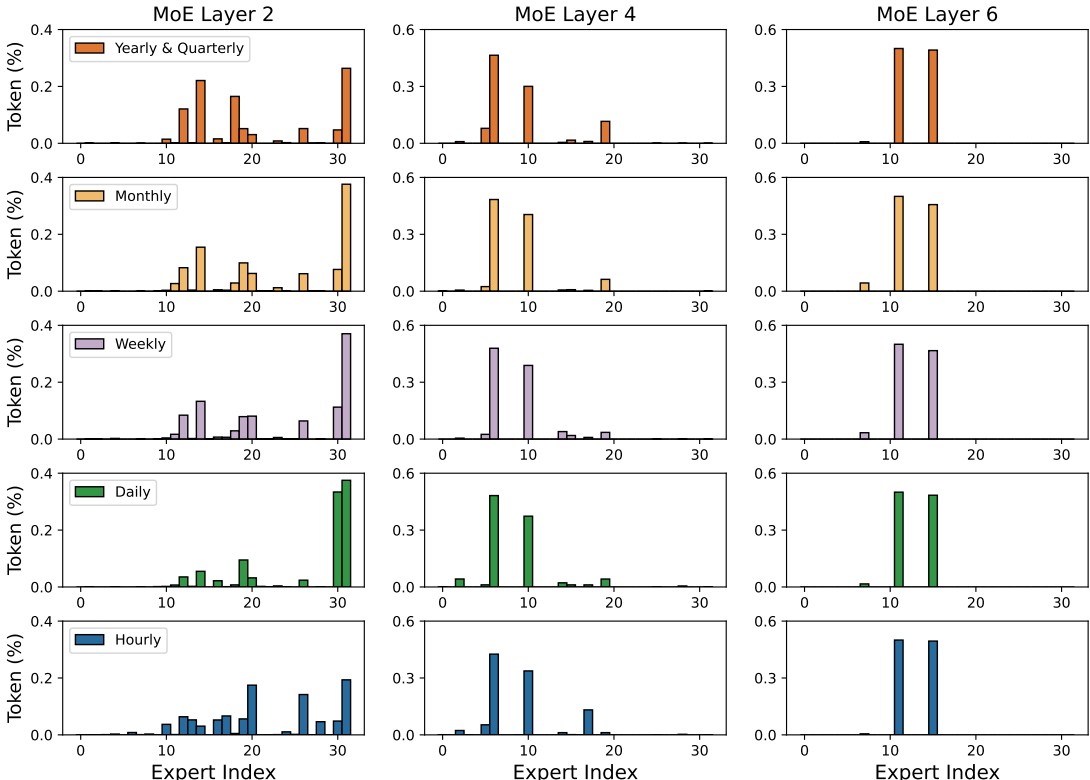

*Figure 6.* Visualization of the distribution of expert allocation for MOIRAI-MOE$_S$ layers 2, 4, and 6 (the last layer) using the Monash benchmark grouped by time series frequency.

*Table 4.* Inference cost evaluation. The values in brackets represent the parameter sizes of the foundation models. For MoE models, the two values indicate the number of activated parameters and the total number of parameters. The spent time is in seconds.

| Model | Chronos$_S$ (46M) | Chronos$_B$ (200M) | Chronos$_L$ (710M) | MOIRAI$_S$ (14M) | MOIRAI$_B$ (91M) | MOIRAI$_L$ (310M) | MOIRAI-MOE$_S$ (11M/117M) | MOIRAI-MOE$_B$ (86M/935M) |
|---|---|---|---|---|---|---|---|---|
| Spent Time (s) | 551 | 1,177 | 2,780 | 264 | 358 | 537 | 273 | 370 |

## 4.5. Efficiency Analyses

In this section, we aim to validate whether the inference speeds of MOIRAI and MOIRAI-MOE are comparable, as we have configured them with similar activated parameters. Additionally, due to the difference in the inference algorithms (the mask filling in MOIRAI predicts all tokens simultaneously, while the next-token prediction approach in MOIRAI-MOE generates predictions autoregressively), we evaluate the inference cost on a subset of the Monash benchmark where the predicted token is one (corresponding to 16 time steps) to eliminate this discrepancy. To also compare to the foundation model Chronos, we align with the setting in Chronos by setting the context length to 512 and the number of sampling samples to 20.

We present the summarized results in Table 4 and conclude that MOIRAI-MOE$_S$ and MOIRAI-MOE$_B$ exhibit similar inference times to MOIRAI$_S$ and MOIRAI$_B$, respectively. These results highlight that MOIRAI-MOE not only main-

tains the same level of efficiency as MOIRAI but also delivers substantial performance improvements. Additionally, when comparing MOIRAI-MOE to Chronos, which also employs autoregressive inference, we find that MOIRAI-MOE is significantly faster. This speed advantage stems from the fact that MOIRAI-MOE generates predictions using patches of size 16, while Chronos uses a patch size of 1, which greatly affects its inference efficiency.

## 5. Conclusion

In this work, we introduce a pioneering time series MoE foundation model MOIRAI-MOE that utilizes sparse experts to model diverse time series patterns in a data-driven manner. Empirical experiments demonstrate that MOIRAI-MOE not only achieves significant performance improvements over all sizes of its predecessor MOIRAI, but also outperforms other competitive foundation models with much fewer activated parameters. Moreover, comprehensive model analyses

have been conducted to gain a deeper understanding of time series MoE foundation models.

## Impact Statement

This paper presents work whose goal is to advance the field of Machine Learning. There are many potential societal consequences of our work, none which we feel must be specifically highlighted here.

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

# A. Experimental Details

## A.1. In-distribution Forecasting Datasets and Full Performance Results

Following MOIRAI (Woo et al., 2024), we perform evaluations on 29 datasets from the Monash benchmark (Godahewa et al., 2021), including M1 Monthly, M3 Monthly, M3 Other, M4 Monthly, M4 Weekly, M4 Daily, M4 Hourly, Tourism Quarterly, Tourism Monthly, CIF 2016, Australian Electricity Demand, Bitcoin, Pedestrian Counts, Vehicle Trips, KDD Cup 2018, Australia Weather, NN5 Daily, NN5 Weekly, Carparts, FRED-MD, Traffic Hourly, Traffic Weekly, Rideshare, Hospital, COVID Deaths, Temperature Rain, Sunspot, Saugeen River Flow, and US Births. The statistics of data are provided in Table 5 and full results of time series foundation models are shown in Table 6.

*Table 5.* Summary of datasets used in the in-distribution forecasting evaluations.

| Dataset | Domain | Frequency | Number of Series | Prediction Length |
|---|---|---|---|---|
| M1 Monthly | Econ/Fin | M | 617 | 18 |
| M3 Monthly | Econ/Fin | M | 1,428 | 18 |
| M3 Other | Econ/Fin | M | 174 | 8 |
| M4 Monthly | Econ/Fin | M | 48,000 | 18 |
| M4 Weekly | Econ/Fin | W | 359 | 13 |
| M4 Daily | Econ/Fin | D | 4,227 | 14 |
| M4 Hourly | Econ/Fin | H | 414 | 48 |
| Tourism Quarterly | Econ/Fin | Q | 427 | 8 |
| Tourism Monthly | Econ/Fin | M | 366 | 24 |
| CIF 2016 | Econ/Fin | M | 72 | 12 |
| Aus. Elec. Demand | Energy | 30T | 5 | 336 |
| Bitcoin | Econ/Fin | D | 18 | 30 |
| Pedestrain Counts | Transport | H | 66 | 24 |
| Vehicle Trips | Transport | D | 329 | 30 |
| KDD Cup 2018 | Energy | H | 270 | 168 |
| Australia Weather | Nature | D | 3,010 | 30 |
| NN5 Daily | Econ/Fin | D | 111 | 56 |
| NN5 Weekly | Econ/Fin | W | 111 | 8 |
| Carparts | Sales | M | 2,674 | 12 |
| FRED-MD | Econ/Fin | M | 107 | 12 |
| Traffic Hourly | Transport | H | 862 | 168 |
| Traffic Weekly | Transport | W | 862 | 8 |
| Rideshare | Transport | H | 2,304 | 168 |
| Hospital | Healthcare | M | 767 | 12 |
| COVID Deaths | Healthcare | D | 266 | 30 |
| Temperature Rain | Nature | D | 32,072 | 30 |
| Sunspot | Nature | D | 1 | 30 |
| Saugeen River Flow | Nature | D | 1 | 30 |
| US Births | Healthcare | D | 1 | 30 |

## A.2. Zero-shot Forecasting Datasets and Full Performance Results

**Dataset Details**  We conduct zero-shot evaluations on the datasets listed in Table 7, which cover five domains and span frequencies ranging from minute-level to weekly. We use a non-overlapping rolling window approach, where the stride equals the prediction length. The test set consists of the last $h * r$ time steps, where $h$ is the forecast horizon and $r$ is the number of rolling evaluation windows. The validation set is defined as the last forecast horizon before the test set, while the training set includes all preceding data.

**Dataset Selection Justification**  The benchmarks set by previous studies – specifically the datasets ETTh1 (H), ETTh2 (H), ETTm1 (15T), ETTm2 (15T), Electricity (H), Weather (10T), Traffic (H), and Solar (10T) – exhibit a strong bias toward the electricity data and the frequencies of hourly and minutely. Additionally, ETTh1 and ETTm1, as well as ETTh2 and ETTm2, are essentially derived from the same underlying data. Furthermore, some of these datasets were already part of our pretraining corpus, raising potential data leakage concerns.

*Table 6.* Full MAE results of time series foundation models on the Monash Benchmark. Other baseline results are in (Woo et al., 2024).

| Dataset | Seasonal Naive | LLMTime | TimesFM | MOIRAI$_{Small}$ | MOIRAI$_{Base}$ | MOIRAI$_{Large}$ | Chronos$_{Small}$ | Chronos$_{Base}$ | Chronos$_{Large}$ | MOIRAI-MOE$_{Small}$ | MOIRAI-MOE$_{Base}$ |
|---|---|---|---|---|---|---|---|---|---|---|---|
| M1 Monthly | 2,011.96 | 2,562.84 | 1,673.60 | 2,082.26 | 2,068.63 | 1,983.18 | 1,797.78 | 1,637.68 | 1,627.11 | 1,992.49 | 1,811.94 |
| M3 Monthly | 788.95 | 877.97 | 653.57 | 713.41 | 658.17 | 664.03 | 644.38 | 622.27 | 619.79 | 646.07 | 617.31 |
| M3 Other | 375.13 | 300.30 | 207.23 | 263.54 | 198.62 | 202.41 | 196.59 | 191.80 | 205.93 | 185.89 | 179.92 |
| M4 Monthly | 700.24 | 728.27 | 580.20 | 597.60 | 592.09 | 584.36 | 592.85 | 598.46 | 584.78 | 569.25 | 544.08 |
| M4 Weekly | 347.99 | 518.44 | 285.89 | 339.76 | 328.08 | 301.52 | 264.56 | 252.26 | 248.89 | 302.65 | 278.37 |
| M4 Daily | 180.83 | 266.52 | 172.98 | 189.10 | 192.66 | 189.78 | 169.91 | 177.49 | 168.41 | 172.45 | 163.40 |
| M4 Hourly | 353.86 | 576.06 | 196.20 | 268.04 | 209.87 | 197.79 | 214.18 | 230.70 | 201.14 | 241.58 | 217.35 |
| Tourism Quarterly | 11,405.45 | 16,918.86 | 10,568.92 | 18,352.44 | 17,196.86 | 15,820.02 | 7,823.27 | 8,835.52 | 8,521.70 | 9,508.07 | 7,374.27 |
| Tourism Monthly | 1,980.21 | 5,608.61 | 2,422.01 | 3,569.85 | 2,862.06 | 2,688.55 | 2,465.10 | 2,358.67 | 2,140.73 | 2,523.66 | 2,268.31 |
| CIF 2016 | 743,512.31 | 599,313.84 | 819,922.44 | 655,888.58 | 539,222.03 | 695,156.92 | 649,110.99 | 604,088.54 | 728,981.15 | 453,631.21 | 568,283.48 |
| Aus. Elec. Demand | 455.96 | 760.81 | 525.73 | 266.57 | 201.39 | 177.68 | 267.18 | 236.27 | 330.04 | 215.28 | 227.92 |
| Bitcoin | 7.78E+17 | 1.74E+18 | 7.78E+17 | 1.76E+18 | 1.62E+18 | 1.87E+18 | 2.34E+18 | 2.27E+18 | 1.88E+18 | 1.55E+18 | 1.90E+18 |
| Pedestrian Counts | 65.60 | 97.77 | 45.03 | 54.88 | 54.08 | 41.66 | 29.77 | 27.34 | 26.95 | 41.35 | 32.37 |
| Vehicle Trips | 32.48 | 31.48 | 21.93 | 24.46 | 23.17 | 21.85 | 19.38 | 19.25 | 19.19 | 21.62 | 21.65 |
| KDD Cup 2018 | 47.09 | 42.72 | 40.86 | 39.81 | 38.66 | 39.09 | 38.60 | 42.36 | 38.83 | 40.21 | 40.86 |
| Australia Weather | 2.36 | 2.17 | 2.07 | 1.96 | 1.80 | 1.75 | 1.96 | 1.84 | 1.85 | 1.76 | 1.75 |
| NN5 Daily | 8.26 | 7.10 | 3.85 | 5.37 | 4.26 | 3.77 | 3.83 | 3.67 | 3.53 | 4.04 | 3.49 |
| NN5 Weekly | 16.71 | 15.76 | 15.09 | 15.07 | 16.42 | 15.30 | 15.03 | 15.12 | 15.09 | 15.74 | 15.29 |
| Carparts | 0.67 | 0.44 | 0.50 | 0.53 | 0.47 | 0.49 | 0.52 | 0.54 | 0.53 | 0.45 | 0.44 |
| FRED-MD | 5,385.53 | 2,804.64 | 2,237.63 | 2,568.48 | 2,679.29 | 2,792.55 | 938.46 | 1,036.67 | 863.99 | 1,651.76 | 2,273.61 |
| Traffic Hourly | 0.013 | 0.030 | 0.009 | 0.020 | 0.020 | 0.010 | 0.013 | 0.012 | 0.010 | 0.013 | 0.014 |
| Traffic Weekly | 1.19 | 1.15 | 1.06 | 1.17 | 1.14 | 1.13 | 1.14 | 1.12 | 1.12 | 1.13 | 1.14 |
| Rideshare | 1.60 | 6.28 | 1.36 | 1.35 | 1.39 | 1.29 | 1.27 | 1.33 | 1.30 | 1.26 | 1.26 |
| Hospital | 20.01 | 25.68 | 18.54 | 23.00 | 19.40 | 19.44 | 19.74 | 19.75 | 19.88 | 20.17 | 19.60 |
| COVID Deaths | 353.71 | 653.31 | 623.47 | 124.32 | 126.11 | 117.11 | 207.47 | 118.26 | 190.01 | 119.00 | 102.92 |
| Temperature Rain | 9.39 | 6.37 | 5.27 | 5.30 | 5.08 | 5.27 | 5.35 | 5.17 | 5.19 | 5.33 | 5.36 |
| Sunspot | 3.93 | 5.07 | 1.07 | 0.11 | 0.08 | 0.13 | 0.20 | 2.45 | 3.45 | 0.10 | 0.08 |
| Saugeen River Flow | 21.50 | 34.84 | 25.16 | 24.07 | 24.40 | 24.76 | 23.57 | 25.54 | 26.25 | 23.05 | 24.40 |
| US Births | 1,152.67 | 1,374.99 | 461.58 | 872.51 | 624.30 | 476.50 | 432.14 | 420.08 | 432.14 | 411.61 | 385.24 |

To address these issues and establish a more diverse zero-shot evaluation scenario without data leakage, we adopt six zero-shot datasets from the MOIRAI paper and incorporate four additional datasets. These added datasets enhance the diversity of our evaluation, making our evaluation cover five domains and include frequencies ranging from minute-level to weekly. However, due to the inherent nature of some datasets, such as limited test set lengths, it is not feasible to apply the same prediction length uniformly across all data. Consequently, the prediction lengths vary across datasets in this study.

**Full Performance Results Measured by 6 Metrics** To comprehensively evaluate performance, we use three point forecasting metrics: mean absolute scaled error (MASE), mean squared error (MSE), and mean absolute error (MAE). For probabilistic forecasts, we evaluate using continuous ranked probability score (CRPS), prediction interval coverage probability (PICP) (Yao et al., 2019), and quantile interval coverage error (QICE) (Han et al., 2022).

Performance results are presented in Table 8, 9, and 10. The results show that MOIRAI-MOE achieves the best overall performance across all 6 metrics, highlighting its superiority in modeling point forecast accuracy, distribution accuracy and distribution coverage.

*Table 7.* Summary of datasets used in the zero-shot forecasting evaluations.

| Dataset | Domain | Frequency | Prediction Length | Rolling Evaluations |
|---|---|---|---|---|
| Electricity (Trindade, 2015) | Energy | H | 24 | 7 |
| Solar (Lai et al., 2018) | Energy | H | 24 | 7 |
| Turkey Power [1] | Energy | H | 24 | 7 |
| ETT1 (Zhou et al., 2021) | Energy | D | 30 | 3 |
| ETT2 (Zhou et al., 2021) | Energy | D | 30 | 3 |
| Istanbul Traffic [2] | Transport | H | 24 | 7 |
| M-DENSE (Jiang et al., 2023) | Transport | D | 30 | 3 |
| Walmart (Walmart Competition Admin, 2014) | Sales | W | 8 | 4 |
| Jena Weather (Wu et al., 2021) | Nature | 10T | 144 | 7 |
| BizITObs-L2C (Palaskar et al., 2024) | Web/CloudOps | 5T | 48 | 20 |

---

[1]https://www.kaggle.com/datasets/dharanikra/electrical-power-demand-in-turkey

[2]https://www.kaggle.com/datasets/leonardo00/istanbul-traffic-index

*Table 8.* Full zero-shot performance comparisons measured by CRPS and MASE. **\*** indicates the non-zero-shot datasets. The Avg column is computed by normalizing each dataset's MAE by seasonal naive, then applying the geometric mean. Two Avg values are shown: one that averages all, and another (non-leak) excludes Electricity and Solar. Best results are in red. Second best results are in blue. Power: Turkey Power. Traffic: Istanbul Traffic. Weather: Jena Weather. BizITObs: BizITObs-L2C.

| Method | Metric | Electricity | Solar | Power | ETT1 | ETT2 | Traffic | MDENSE | Walmart | Weather | BizITObs | Avg (all) | Avg (non-leak) |
|---|---|---|---|---|---|---|---|---|---|---|---|---|---|
| Seasonal Naive | CRPS | 0.070 | 0.512 | 0.085 | 0.515 | 0.205 | 0.257 | 0.294 | 0.151 | 0.068 | 0.262 | 1.000 | 1.000 |
| | MASE | 0.881 | 1.203 | 0.906 | 1.778 | 1.390 | 1.137 | 1.669 | 1.236 | 0.782 | 0.986 | 1.000 | 1.000 |
| TiDE | CRPS | 0.048 | 0.420 | 0.046 | 1.056 | 0.130 | 0.110 | 0.091 | 0.077 | 0.054 | 0.124 | 0.631 | 0.604 |
| | MASE | 0.706 | 1.265 | 0.904 | 6.898 | 2.189 | 0.618 | 0.911 | 0.814 | 0.832 | 0.450 | 0.931 | 0.934 |
| PatchTST | CRPS | 0.052 | 0.518 | 0.054 | 0.304 | 0.131 | 0.112 | 0.070 | 0.082 | 0.059 | 0.074 | 0.549 | 0.490 |
| | MASE | 0.753 | 1.607 | 1.234 | 1.680 | 2.168 | 0.653 | 0.732 | 0.867 | 0.844 | 0.266 | 0.808 | 0.753 |
| iTransformer | CRPS | 0.057 | 0.443 | 0.056 | 0.344 | 0.129 | 0.105 | 0.072 | 0.070 | 0.053 | 0.077 | 0.540 | 0.483 |
| | MASE | 0.875 | 1.342 | 1.076 | 2.393 | 1.841 | 0.581 | 0.727 | 0.761 | 0.623 | 0.271 | 0.767 | 0.708 |
| MoLE-DLinear | CRPS | 0.083 | 0.535 | 0.072 | 0.344 | 0.188 | 0.237 | 0.108 | 0.137 | 0.079 | 0.095 | 0.780 | 0.714 |
| | MASE | 0.984 | 1.257 | 1.325 | 1.606 | 3.194 | 1.016 | 0.914 | 1.115 | 0.925 | 0.282 | 0.938 | 0.906 |
| TTM | CRPS | 0.075 | 0.534* | 0.059 | 0.417 | 0.122 | 0.210 | 0.150 | 0.192 | 0.055 | 0.102 | 0.758 | 0.697 |
| | MASE | 0.802 | 1.255* | 0.898 | 1.934 | 1.547 | 0.901 | 1.195 | 1.477 | 0.506 | 0.308 | 0.831 | 0.798 |
| Timer | CRPS | 0.084 | 0.573 | 0.066 | 0.345 | 0.135 | 0.182 | 0.152 | 0.151 | 0.092 | 0.120 | 0.797 | 0.726 |
| | MASE | 0.967 | 1.344 | 1.006 | 1.697 | 1.754 | 0.770 | 1.196 | 1.219 | 0.655 | 0.376 | 0.871 | 0.820 |
| TimesFM | CRPS | 0.045* | 0.456 | 0.037 | 0.280 | 0.113 | 0.131 | 0.070 | 0.067 | 0.042 | 0.080 | 0.488 | 0.439 |
| | MASE | 0.655* | 1.391 | 0.851 | 1.700 | 1.644 | 0.678 | 0.702 | 0.735 | 0.440 | 0.310 | 0.689 | 0.640 |
| Chronos$_S$ | CRPS | 0.043* | 0.389* | 0.038 | 0.360 | 0.097 | 0.124 | 0.087 | 0.079 | 0.089 | 0.087 | 0.543 | 0.513 |
| | MASE | 0.629* | 1.193* | 0.717 | 1.799 | 1.431 | 0.622 | 0.834 | 0.849 | 0.606 | 0.301 | 0.694 | 0.661 |
| Chronos$_B$ | CRPS | 0.041* | 0.341* | 0.039 | 0.387 | 0.092 | 0.109 | 0.075 | 0.080 | 0.058 | 0.084 | 0.499 | 0.471 |
| | MASE | 0.617* | 1.002* | 0.722 | 1.898 | 1.265 | 0.553 | 0.712 | 0.849 | 0.583 | 0.301 | 0.656 | 0.631 |
| Chronos$_L$ | CRPS | 0.041* | 0.339* | 0.038 | 0.404 | 0.091 | 0.117 | 0.075 | 0.073 | 0.062 | 0.084 | 0.500 | 0.473 |
| | MASE | 0.615* | 0.987* | 0.702 | 1.959 | 1.270 | 0.597 | 0.724 | 0.788 | 0.601 | 0.310 | 0.660 | 0.638 |
| MOIRAI$_S$ | CRPS | 0.072 | 0.471 | 0.048 | 0.275 | 0.101 | 0.173 | 0.084 | 0.103 | 0.049 | 0.081 | 0.578 | 0.507 |
| | MASE | 0.981 | 1.465 | 0.948 | 1.701 | 1.417 | 0.990 | 0.836 | 1.048 | 0.521 | 0.301 | 0.798 | 0.726 |
| MOIRAI$_B$ | CRPS | 0.055 | 0.419 | 0.040 | 0.301 | 0.095 | 0.116 | 0.104 | 0.093 | 0.041 | 0.078 | 0.520 | 0.467 |
| | MASE | 0.792 | 1.292 | 0.888 | 1.736 | 1.314 | 0.644 | 1.101 | 0.964 | 0.487 | 0.291 | 0.736 | 0.685 |
| MOIRAI$_L$ | CRPS | 0.050 | 0.406 | 0.036 | 0.286 | 0.094 | 0.112 | 0.095 | 0.098 | 0.051 | 0.079 | 0.514 | 0.467 |
| | MASE | 0.751 | 1.237 | 0.870 | 1.750 | 1.436 | 0.631 | 0.957 | 1.007 | 0.515 | 0.285 | 0.729 | 0.685 |
| Time-MoE$_B$ | CRPS | 0.051* | 0.230* | 0.044 | 0.392 | 0.125 | 0.152 | 0.099 | 0.100 | 0.070 | 0.112 | 0.583 | 0.586 |
| | MASE | 0.587* | 0.535* | 0.800 | 1.823 | 1.672 | 0.672 | 0.846 | 0.833 | 0.558 | 0.343 | 0.662 | 0.695 |
| Time-MoE$_L$ | CRPS | 0.051* | 0.294* | 0.045 | 0.386 | 0.131 | 0.172 | 0.090 | 0.097 | 0.058 | 0.111 | 0.589 | 0.576 |
| | MASE | 0.581* | 0.689* | 0.790 | 1.773 | 1.878 | 0.762 | 0.759 | 0.817 | 0.524 | 0.337 | 0.678 | 0.695 |
| MOIRAI-MoE$_S$ | CRPS | 0.046 | 0.429 | 0.036 | 0.288 | 0.093 | 0.108 | 0.071 | 0.090 | 0.056 | 0.081 | 0.497 | 0.450 |
| | MASE | 0.719 | 1.222 | 0.737 | 1.750 | 1.248 | 0.563 | 0.746 | 0.927 | 0.476 | 0.298 | 0.670 | 0.620 |
| MOIRAI-MoE$_B$ | CRPS | 0.041 | 0.382 | 0.034 | 0.296 | 0.091 | 0.100 | 0.071 | 0.088 | 0.057 | 0.079 | 0.478 | 0.439 |
| | MASE | 0.638 | 1.161 | 0.725 | 1.748 | 1.247 | 0.510 | 0.721 | 0.918 | 0.509 | 0.290 | 0.651 | 0.611 |

*Table 9.* Full zero-shot performance comparisons measured by MSE and MAE. **Note that the MSE and MAE values are relatively large compared to those reported in previous studies like PatchTST, primarily because we compute the loss using raw time series values rather than normalized ones. This approach can more accurately reflect the forecasting accuracy gap.** Best results are in red. Second best results are in blue. Power: Turkey Power. Traffic: Istanbul Traffic. Weather: Jena Weather. BizITObs: BizITObs-L2C.

| Method | Metric | Electricity | Solar | Power | ETT1 | ETT2 | Traffic | MDENSE | Walmart | Weather | BizITObs | Avg (all) | Avg (non-leak) |
|---|---|---|---|---|---|---|---|---|---|---|---|---|---|
| Seasonal Naive | MSE | 1299429.16 | 1293.24 | 1798196.83 | 57976.63 | 122878.95 | 203.32 | 39929.67 | 32876026.66 | 2197.23 | 174.31 | 1.000 | 1.000 |
| | MAE | 166.20 | 15.77 | 492.60 | 154.98 | 211.56 | 8.72 | 118.38 | 2637.43 | 10.96 | 9.69 | 1.000 | 1.000 |
| iTransformer | MSE | 1264494.38 | 1183.57 | 968959.56 | 55320.57 | 178757.02 | 41.77 | 9905.39 | 10922819.00 | 1885.01 | 20.55 | 0.508 | 0.435 |
| | MAE | 165.89 | 17.61 | 399.09 | 170.83 | 279.21 | 4.85 | 51.06 | 1560.68 | 10.65 | 2.66 | 0.741 | 0.678 |
| MoLE-DLinear | MSE | 1901617.97 | 1098.56 | 1071490.46 | 39026.37 | 195287.19 | 153.71 | 13016.78 | 26832049.08 | 1649.90 | 21.57 | 0.656 | 0.575 |
| | MAE | 197.06 | 16.47 | 420.67 | 130.79 | 328.28 | 8.48 | 62.43 | 2395.50 | 12.81 | 2.75 | 0.857 | 0.803 |
| TTM | MSE | 2432897.66 | 884.33* | 647289.67 | 56256.46 | 116203.30 | 114.79 | 18425.62 | 39297380.00 | 1122.55 | 23.41 | 0.625 | 0.538 |
| | MAE | 179.56 | 16.46* | 341.96 | 158.85 | 213.61 | 7.53 | 86.44 | 3360.79 | 8.88 | 2.97 | 0.833 | 0.784 |
| Timer | MSE | 2205084.30 | 962.26 | 687600.25 | 39235.36 | 129063.67 | 75.23 | 19875.60 | 29410540.00 | 1873.68 | 27.21 | 0.613 | 0.527 |
| | MAE | 200.62 | 17.57 | 370.53 | 131.31 | 235.27 | 6.42 | 87.72 | 2646.92 | 13.65 | 3.50 | 0.865 | 0.804 |
| TimesFM | MSE | 1378828.95* | 1061.70 | 384815.80 | 42789.02 | 169714.41 | 106.01 | 10194.73 | 9494507.86 | 1317.09 | 23.23 | 0.475 | 0.401 |
| | MAE | 137.57* | 18.07 | 277.94 | 138.42 | 245.61 | 5.75 | 49.78 | 1484.68 | 7.94 | 2.89 | 0.672 | 0.612 |
| Chronos$_S$ | MSE | 1251170.49* | 1405.10* | 418195.72 | 60157.02 | 112472.02 | 100.62 | 15377.29 | 14697271.28 | 3945.04 | 23.89 | 0.587 | 0.511 |
| | MAE | 126.25* | 15.79* | 275.11 | 161.23 | 207.11 | 5.28 | 59.26 | 1693.33 | 16.90 | 2.94 | 0.724 | 0.691 |
| Chronos$_B$ | MSE | 1147348.35* | 1062.73* | 400709.37 | 66320.26 | 107178.21 | 80.48 | 12770.66 | 15813384.14 | 1720.53 | 22.78 | 0.501 | 0.439 |
| | MAE | 121.69* | 13.18* | 285.79 | 169.60 | 194.70 | 4.69 | 51.58 | 1706.11 | 10.28 | 2.82 | 0.656 | 0.628 |
| Chronos$_L$ | MSE | 1073679.39* | 1017.98* | 362386.33 | 73974.48 | 106362.90 | 98.20 | 13625.07 | 12339319.84 | 1874.83 | 23.61 | 0.503 | 0.447 |
| | MAE | 121.06* | 12.86* | 277.64 | 177.68 | 191.07 | 5.07 | 53.61 | 1560.11 | 11.30 | 2.89 | 0.664 | 0.639 |
| MOIRAI$_S$ | MSE | 4015423.50 | 1429.82 | 757613.06 | 39481.46 | 118636.33 | 146.24 | 11041.41 | 19886286.00 | 1932.16 | 22.48 | 0.647 | 0.498 |
| | MAE | 219.02 | 19.19 | 358.01 | 133.82 | 209.68 | 8.71 | 58.25 | 2112.07 | 10.23 | 2.90 | 0.802 | 0.715 |
| MOIRAI$_B$ | MSE | 1734656.25 | 1105.95 | 477193.47 | 51793.64 | 113074.23 | 44.60 | 17724.71 | 18981036.00 | 1196.21 | 22.44 | 0.500 | 0.414 |
| | MAE | 164.94 | 16.97 | 293.74 | 149.15 | 202.89 | 4.72 | 79.41 | 2046.22 | 7.73 | 2.81 | 0.713 | 0.650 |
| MOIRAI$_L$ | MSE | 1229872.00 | 997.13 | 340307.44 | 44752.48 | 106513.38 | 101.17 | 14874.89 | 21274060.00 | 1914.39 | 21.79 | 0.511 | 0.449 |
| | MAE | 150.66 | 16.25 | 262.70 | 142.21 | 204.72 | 5.93 | 69.73 | 2110.73 | 10.10 | 2.77 | 0.720 | 0.669 |
| Time-MoE$_B$ | MSE | 1158323.38* | 176.27* | 315704.91 | 50267.22 | 114374.42 | 89.87 | 11303.31 | 13934856.92 | 1371.87 | 28.51 | 0.395 | 0.408 |
| | MAE | 120.52* | 7.07* | 254.28 | 149.21 | 218.55 | 5.70 | 57.43 | 1742.96 | 11.35 | 3.26 | 0.644 | 0.663 |
| Time-MoE$_L$ | MSE | 1203643.75* | 194.84* | 350989.67 | 47389.70 | 121112.59 | 99.13 | 9585.73 | 12876789.32 | 1264.26 | 27.34 | 0.394 | 0.400 |
| | MAE | 120.53* | 9.06* | 262.48 | 147.11 | 229.67 | 6.45 | 52.10 | 1687.08 | 9.32 | 3.24 | 0.650 | 0.652 |
| MOIRAI-MoE$_S$ | MSE | 930140.63 | 1113.50 | 360995.68 | 45412.81 | 114609.09 | 53.05 | 9426.45 | 18025986.00 | 1944.27 | 23.45 | 0.453 | 0.395 |
| | MAE | 138.03 | 16.05 | 260.82 | 141.08 | 194.63 | 4.78 | 50.09 | 1955.77 | 10.08 | 2.89 | 0.668 | 0.617 |
| MOIRAI-MoE$_B$ | MSE | 907276.31 | 1047.63 | 311227.06 | 48487.21 | 107284.42 | 45.83 | 9740.51 | 17094764.00 | 1954.24 | 22.54 | 0.434 | 0.378 |
| | MAE | 122.27 | 15.24 | 251.10 | 145.50 | 191.47 | 4.33 | 49.73 | 1919.31 | 10.31 | 2.80 | 0.646 | 0.605 |

*Table 10.* Full zero-shot performance comparisons measured by PICP and QICE. The absence of certain baselines reflects their inability to compute PICP and QICE, as they only support point forecasting. The Avg column is computed by taking the geometric mean across all datasets. Best results are in red. Power: Turkey Power. Traffic: Istanbul Traffic. Weather: Jena Weather. BizITObs: BizITObs-L2C.

| Method | Metric | Electricity | Solar | Power | ETT1 | ETT2 | Traffic | MDENSE | Walmart | Weather | BizITObs | Avg (all) |
|---|---|---|---|---|---|---|---|---|---|---|---|---|
| Chronos$_S$ | PICP | 8.653 | 55.871 | 21.356 | 39.286 | 5.635 | 21.720 | 13.000 | 17.612 | 43.276 | 21.042 | 20.021 |
| | QICE | 1.694 | 6.192 | 3.144 | 4.885 | 1.129 | 3.857 | 2.189 | 2.341 | 5.331 | 3.480 | 3.037 |
| Chronos$_B$ | PICP | 8.201 | 56.031 | 23.075 | 46.111 | 4.841 | 18.280 | 11.333 | 17.494 | 44.140 | 18.021 | 19.205 |
| | QICE | 1.785 | 6.205 | 3.490 | 5.697 | 1.869 | 3.545 | 2.305 | 2.298 | 5.335 | 2.887 | 3.218 |
| Chronos$_L$ | PICP | 7.645 | 56.379 | 20.827 | 43.095 | 10.079 | 14.312 | 8.889 | 16.530 | 43.110 | 17.485 | 19.015 |
| | QICE | 1.661 | 6.243 | 3.373 | 5.467 | 1.746 | 3.868 | 1.901 | 2.216 | 5.253 | 2.708 | 3.082 |
| MOIRAI$_S$ | PICP | 1.184 | 3.436 | 1.283 | 1.508 | 1.984 | 3.413 | 0.259 | 1.908 | 3.239 | 3.452 | 1.766 |
| | QICE | 0.915 | 2.152 | 2.786 | 2.698 | 2.610 | 2.769 | 2.342 | 1.508 | 1.433 | 0.971 | 1.871 |
| MOIRAI$_B$ | PICP | 1.132 | 3.549 | 0.470 | 1.508 | 3.730 | 2.884 | 0.148 | 1.416 | 0.267 | 2.098 | 1.137 |
| | QICE | 0.422 | 1.469 | 1.372 | 2.575 | 2.698 | 3.504 | 3.140 | 1.100 | 1.196 | 0.997 | 1.560 |
| MOIRAI$_L$ | PICP | 0.032 | 2.545 | 0.489 | 1.984 | 3.254 | 1.296 | 1.778 | 1.385 | 0.588 | 5.223 | 1.098 |
| | QICE | 0.343 | 3.506 | 1.357 | 2.504 | 1.834 | 3.063 | 1.140 | 0.827 | 1.271 | 2.396 | 1.520 |
| MOIRAI-MoE$_S$ | PICP | 0.717 | 4.150 | 1.515 | 0.873 | 3.254 | 5.582 | 0.556 | 2.957 | 1.452 | 1.042 | 1.678 |
| | QICE | 0.899 | 1.756 | 0.979 | 2.222 | 2.769 | 1.952 | 0.819 | 1.082 | 2.157 | 0.931 | 1.419 |
| MOIRAI-MoE$_B$ | PICP | 0.464 | 0.473 | 0.172 | 0.397 | 4.206 | 9.550 | 0.259 | 3.118 | 1.528 | 2.173 | 1.049 |
| | QICE | 0.712 | 1.174 | 1.703 | 2.310 | 2.628 | 2.734 | 3.041 | 1.158 | 1.601 | 0.908 | 1.615 |

## A.3. Summary of Methods

The following is a brief introduction to the models used in the evaluation process.

- TiDE (Das et al., 2023) encodes the historical data of a time series along with covariates using dense multi-layer perceptrons (MLPs). It then decodes the time series while incorporating future covariates, also utilizing dense MLPs for this process.

- PatchTST (Nie et al., 2023) employs Transformer encoders combined with patching and channel independence techniques to enhance the performance of time series forecasting.

- iTransformer (Liu et al., 2024b) treats independent time series as tokens to effectively capture multivariate correlations through self-attention.

- MoLE-DLinear (Ni et al., 2024) trains multiple linear-centric models (i.e., experts) and a router model that weighs and mixes their outputs. In this study, we use the DLinear model as the experts.

- LLMTime (Gruver et al., 2023) is a method for time series forecasting that leverages Large Language Models by encoding numerical data as text and generating possible future values through text completions.

- Moment (Goswami et al., 2024) refers to a family of open time series foundation models that canhandle different time series analysis tasks. **Note that Moment requires fine-tuning for forecast tasks and cannot directly do zero-shot forecasting, as stated in the GitHub issue:** `https://github.com/moment-timeseries-foundation-model/moment/issues/21#issuecomment-2138478827`. **Therefore, we exclude it in our model comparisons.**

- TTM (Ekambaram et al., 2024) is a foundation model based on the light-weight TSMixer architecture, incorporating innovations like adaptive patching, diverse resolution sampling, and resolution prefix tuning.

- Timer (Liu et al., 2024c) is a decoder-only foundation model, presenting notable few-shot generalization, scalability, and task generality.

- TimesFM (Das et al., 2024) is a decoder-only time series foundation model that pretrained on a large corpus of time series data, including both real-world and synthetic datasets.

- Chronos (Ansari et al., 2024) is an encoder-decoder time series foundation model that uses quantization to convert real numbers into discrete tokens.

- MOIRAI (Woo et al., 2024) is a time series foundation model trained on the LOTSA dataset, which contains over 27 billion observations across nine diverse domains.

- Time-MoE (Shi et al., 2024) is a concurrent work that applies mixture of experts techniques to time series foundation models.

- MOIRAI-MOE is proposed in this study, which is capable of achieving automatic token-level specialization.

## A.4. Settings of Methods

**Context Length Setting for All Methods**    In Table 11, we detail the context lengths used for each method in this study, and in their original paper. For full-shot deep learning models, we believe our searching range generally covers the lengths set in their original paper. For foundation models, the choice of input lengths depends on their pretraining strategies. For instance, in the case of TimesFM and Chronos, the input lengths are consistently set to 512 during pretraining. In contrast, for MOIRAI and MOIRAI-MOE, the pretraining algorithm involves randomly sampling a context length in the range [2, 8192]. Thus, searching for the input length on validation set during inference is needed.

*Table 11.* Comparison of methods' context lengths: this study versus original papers.

| Model | In-Dist. Evaluation (29 datasets) | Zero-Shot Evaluation (10 datasets) | Original Paper |
|---|---|---|---|
| TiDE | – | Searching within prediction lengths * [2,20] | 720 |
| PatchTST | – | Searching within prediction lengths * [2,20] | 336 |
| iTransformer | – | Searching within prediction lengths * [2,20] | 96 |
| TTM | – | 512 | 512 |
| Timer | – | 672 | 672 |
| Time-MoE | – | 4,096 | {512, 1024, 2048, 3072} |
| TimesFM | 512 | 512 | 512 |
| Chronos | 512 | 512 | 512 |
| MOIRAI | 1000 | Searching within range {1000, 2000, 3000, 4000, 5000} | Searching within range {1000, 2000, 3000, 4000, 5000} |
| MOIRAI-MOE | 1000 | Searching within range {1000, 2000, 3000, 4000, 5000} | Searching within range {1000, 2000, 3000, 4000, 5000} |

**Long Context Lengths for Full-Shot methods**    To investigate the potential unfair comparisons arising from relatively shot context lengths of full-shot methods compared to foundation models, additional experiments are conducted: training the state-of-the-art specialized models PatchTST and iTransformers using the same context length searching strategy as in MOIRAI and MOIRAI-MOE.

Table 12 presents the results using MSE and MAE. Note that, for clarity, we use Range-1 to denote the searching range within prediction lengths * [2, 20], which is approximately around the values of [50, 500]. This is the range we used in our paper for specialized models. In addition, we use Range-2 to denote MOIRAI-MOE's strategy: searching within 1000, 2000, 3000, 4000, 5000. According to the results, we observe that from Range-1 to Range-2, the input length searching range increases substantially. However, the average performance of both PatchTST and iTransformer declines significantly across all four metrics. For instance, the average MASE of PatchTST increases from 0.808 to 0.967, and the average MASE of iTransformer rises from 0.767 to 1.013. These results suggest that neither PatchTST nor iTransformer benefits from very long input lengths; in fact, such lengths negatively impact their performance. And the performance gap between these models and the foundation models becomes even more pronounced.

*Table 12.* Performance comparison for different context lengths measured by MSE and MAE. Best results are in **bold**. Power: Turkey Power. Traffic: Istanbul Traffic. Weather: Jena Weather. BizITObs: BizITObs-L2C.

| Method | Context Length | Metric | Electricity | Solar | Power | ETT1 | ETT2 | Traffic | MDENSE | Walmart | Weather | BizITObs | Avg (all) |
|---|---|---|---|---|---|---|---|---|---|---|---|---|---|
| Seasonal Naive | – | MSE | 1299429.16 | 1293.24 | 1798196.83 | 57976.63 | 122878.95 | 203.32 | 39929.67 | 32876026.66 | 2197.23 | 174.31 | 1.000 |
| | – | MAE | 166.20 | 15.77 | 492.60 | 154.98 | 211.56 | 8.72 | 118.38 | 2637.43 | 10.96 | 9.69 | 1.000 |
| PatchTST | Range-1 | MSE | 1534813.00 | 1125.03 | 1605878.50 | 45755.70 | 167280.13 | 38.69 | 9240.59 | 13749435.00 | 1361.20 | 18.97 | 0.511 |
| | Range-1 | MAE | 171.95 | 17.03 | 478.86 | 141.03 | 291.80 | 4.51 | 49.45 | 1847.65 | 2.60 | 0.740 |
| PatchTST | Range-2 | MSE | 1433466.75 | 973.65 | 957680.50 | 249211.48 | 284311.13 | 45.14 | 38315.62 | 15738720.00 | 2020.57 | 15.18 | 0.717 |
| | Range-2 | MAE | 150.33 | 15.69 | 414.32 | 259.93 | 346.99 | 4.96 | 111.85 | 1859.13 | 15.96 | 2.36 | 0.880 |
| iTransformer | Range-1 | MSE | 1264494.38 | 1183.57 | 968959.56 | 55320.57 | 178757.02 | 41.77 | 9905.39 | 10922819.00 | 1885.01 | 20.55 | 0.508 |
| | Range-1 | MAE | 165.89 | 17.61 | 399.09 | 170.83 | 279.21 | 4.85 | 51.06 | 1560.68 | 10.65 | 2.66 | 0.741 |
| iTransformer | Range-2 | MSE | 1856541.38 | 2587.88 | 572127.56 | 54057.63 | 297574.38 | 39.58 | 19981.01 | 21888656.00 | 1375.14 | 50.39 | 0.689 |
| | Range-2 | MAE | 174.00 | 23.96 | 350.17 | 167.26 | 408.40 | 4.74 | 89.69 | 2504.30 | 10.11 | 5.53 | 0.930 |
| MOIRAI-MOE_B | Range-2 | MSE | 907276.31 | 1047.63 | 311227.06 | 48487.21 | 107284.42 | 45.83 | 9740.51 | 17094764.00 | 1954.24 | 22.54 | **0.434** |
| | Range-2 | MAE | 122.27 | 15.24 | 251.10 | 145.50 | 191.47 | 4.33 | 49.73 | 1919.31 | 10.31 | 2.80 | **0.646** |

**Hyperparameter Search for Full-Shot Methods**    For the three full-shot models used in zero-shot forecasting part, i.e., TiDE (Das et al., 2023), PatchTST (Nie et al., 2023), and iTransformer (Liu et al., 2024b), we conduct hyperparameter search based on the values specified in Table 13. In addition, we explore the learning rate in the range [1e-6, 1e-3] on a log scale, and set the context length as $l = m * h$, where $m$ is tuned in the range [2, 20], and $h$ is the prediction length. We implement a random search across these parameters over 15 runs and report results based on the best validation CRPS.

**Pretraining Details of MOIRAI-MOE**    All MOIRAI-MOE models are trained on 16 A100 (40G) GPUs using a batch size of 1,024 and bfloat16 precision. The small and base model are trained for 50,000 and 250,000 steps on LOTSA (Woo et al., 2024), respectively. The patch size $P$ is set to 16 and the masking ratio $r$ for next-token prediction pretraining is 0.3. The corresponding experiments are in Appendix B.3 and B.4. For optimization, we utilize the AdamW optimizer with lr = 1e-3, weight decay = 1e-1, $\beta_1 = 0.9$, $\beta_2 = 0.98$. We also apply a learning rate scheduler with linear warmup for the first 10,000 steps, followed by cosine annealing.

*Table 13.* Hyperparameter search values for TiDE, PatchTST, and iTransformer.

| | Hyperparameter | Values |
|---|---|---|
| TiDE | hidden_dim | {64, 128, 256} |
| | num_encoder_layers | [2, 6] |
| | num_decoder_layers | [2, 6] |
| PatchTST | d_model | {64, 128, 256} |
| | num_encoder_layers | [2,6] |
| iTransformer | d_model | {128, 256, 512} |
| | num_encoder_layers | [2, 4] |

## B. Additional Results

### B.1. Comparison of MOIRAI and MOIRAI-MOE Pretraining Steps

In Figure 7, we present a comparison between MOIRAI$_S$ and MOIRAI-MOE$_S$ in terms of pretraining steps. The results demonstrate that MOIRAI-MOE$_S$ outperforms MOIRAI$_S$ from the very first evaluation point – 25k steps. Furthermore, MOIRAI-MOE$_S$ at 25k steps achieves better performance than MOIRAI$_S$ at 125k steps. This figure highlights the clear advantages of MOIRAI-MOE in terms of both model performance and reduced pretraining steps.

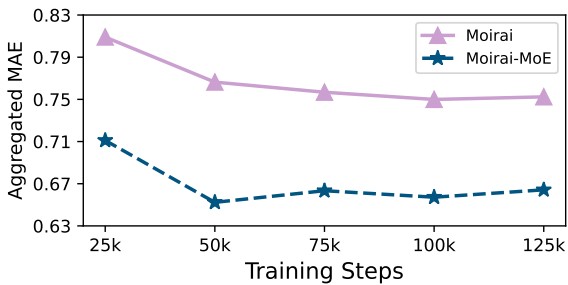

*Figure 7.* Performance comparison between MOIRAI$_S$ and MOIRAI-MOE$_S$ across pretraining steps.

### B.2. Effects of the Number of MoE Layers

According to a recent survey on MoE-based Transformers (Cai et al., 2024), replacing all FFNs with MoE layers is a common practice in recent and well-known LLMs, such as Mixtral 8x7B and Qwen1.5-MoE. Another popular choice is to replace half of the FFN layers with MoE layers. To investigate the efficacy of this, we conduct experiments where one out of every two FFN layers was replaced with MoE layers. As shown in Table 14, compared to replacing all FFN layers, this "half" setting reduces pretraining time by 31%. While the downstream performance on Monash shows that replacing half of the FFN layers results in a 5% performance drop. This demonstrates a trade-off between performance and pretraining cost.

*Table 14.* Effects of the number of MoE layers in MOIRAI-MOE.

| Variant | MoE Layers | Total Layers | Activated Params | Total Params | Monash Performance | Pretraining Time |
|---|---|---|---|---|---|---|
| MOIRAI-MOE$_S$-Half | 3 | 6 | 11M | 64M | 0.68 | 6.53h |
| MOIRAI-MOE$_S$ | 6 | 6 | 11M | 117M | 0.65 | 9.49h |

### B.3. Effects of Patch Size

In contrast to MOIRAI, which designs multiple input/output projection layers, each associated with a specific patch size, MOIRAI-MOE utilizes a single projection layer with a single patch size. In this part, we conduct experiments to examine the impact of different patch size choices. The evaluation results on the Monash benchmark are presented in Figure 8 (left), where the patch size of 16 yields the best performance. Regarding the choice of patch size in relation to performance, the patch size determines the time period encompassed within each token. If the patch size is too large (i.e., 64), the linear projection layer may lack the capacity to capture the underlying patterns. Conversely, if the patch size is too small (i.e., 4),

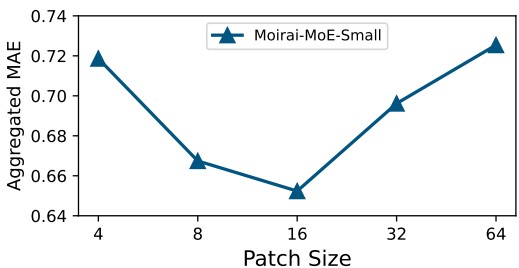 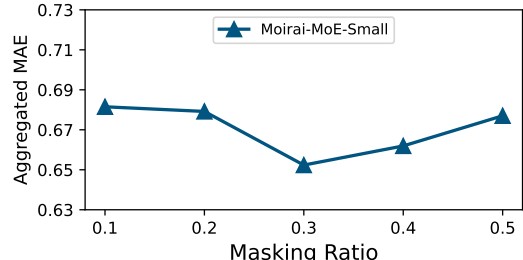

*Figure 8.* Effects of patch size and masking ratio using MOIRAI-MOE$_S$.

the time series token may not contain sufficient semantic information, as highlighted in DLinear. Additionally, patch size affects inference speed; with a fixed context window, smaller patch sizes generate more time series tokens, increasing GPU memory usage and ultimately slowing down inference. For instance, using a patch size of 4 can take over a day to complete all evaluations. Our choice of a patch size of 16 not only delivers strong performance but also maintains a reasonable inference speed.

Furthermore, to justify the choice of a single projection layer over multiple projection layers, we conducted experiments using multiple input/output projection layers within MOIRAI-MOE. The results are presented in Table 15. The notation used is as follows: MOIRAI-MOE$_S$-Multi refers to MOIRAI-MOE$_S$ with a multi-patch strategy. MOIRAI-MOE$_S$-S4 represents MOIRAI-MOE$_S$ with a single patch size of 4. The same naming applies to other configurations. The results show that S4 and S64 do not perform as well as S16, however, they significantly outperform MOIRAI$_S$ and achieve comparable results to MOIRAI-MOE$_S$-Muiti. These results indicate that the multi-patch strategy does not hold positive significance, since it is comparable to the worst cases of single patch size.

*Table 15.* Effects of single projection layer and multiple projection layers in MOIRAI-MOE.

| Method | MOIRAI$_S$ | MOIRAI-MOE$_S$-Multi | MOIRAI-MOE$_S$-S4 | MOIRAI-MOE$_S$-S8 | MOIRAI-MOE$_S$-S16 | MOIRAI-MOE$_S$-S32 | MOIRAI-MOE$_S$-S64 |
|---|---|---|---|---|---|---|---|
| Monash Performance | 0.78 | 0.72 | 0.72 | 0.67 | 0.65 | 0.70 | 0.72 |

### B.4. Effects of Masking Ratio

In this study, we introduce the masking ratio $r$ as a hyperparameter that determines the portion of the entire sequence used solely for robust normalizer calculation, helping to mitigate distribution shift issues. We conduct experiments to assess the effects of different masking ratios, with the evaluation results on the Monash benchmark shown in Figure 8 (right). A masking ratio of 0.3 delivers the best performance. A ratio of 0.1 uses too little data to compute a robust normalizer, potentially failing to accurately represent the overall sequence statistics. Conversely, a ratio of 0.5 masks half of the data, which may hinder the parallel learning efficiency in next-token prediction pretraining. Therefore, it is crucial to select an appropriate data range that is small enough to avoid excessive masking, yet sufficiently representative for robust normalizer computation.

### B.5. Expert Distributions of Different Gating Function

In this part, we present an in-depth comparison of the different gating functions explored in this study. First, we provide additional details on the implementation of the proposed token clustering method. The core idea of this approach is to leverage cluster centroids derived from the token representations of a pretrained model to guide expert allocations. Specifically, we perform inference on our pretraining corpus, `LOTSA`, using data amount corresponding to 100 epochs. During this process, we extract the self-attention output representations from a pretrained MOIRAI model and apply mini-batch k-means clustering to continuously update the clusters. The number of clusters is set to match the total number of experts. During the pretraining of the MoE model, each token computes the Euclidean distance to each cluster centroid, and these distances are used as token-to-expert affinity scores for expert assignments. Empirical evaluations have demonstrated the effectiveness of this approach compared to randomly learned gating from scratch, indicating that the clustering method better aligns with the inherent distribution of time series representations.

Using the three gating functions explored in this study, i.e., linear projection, linear projection with load balancing, and

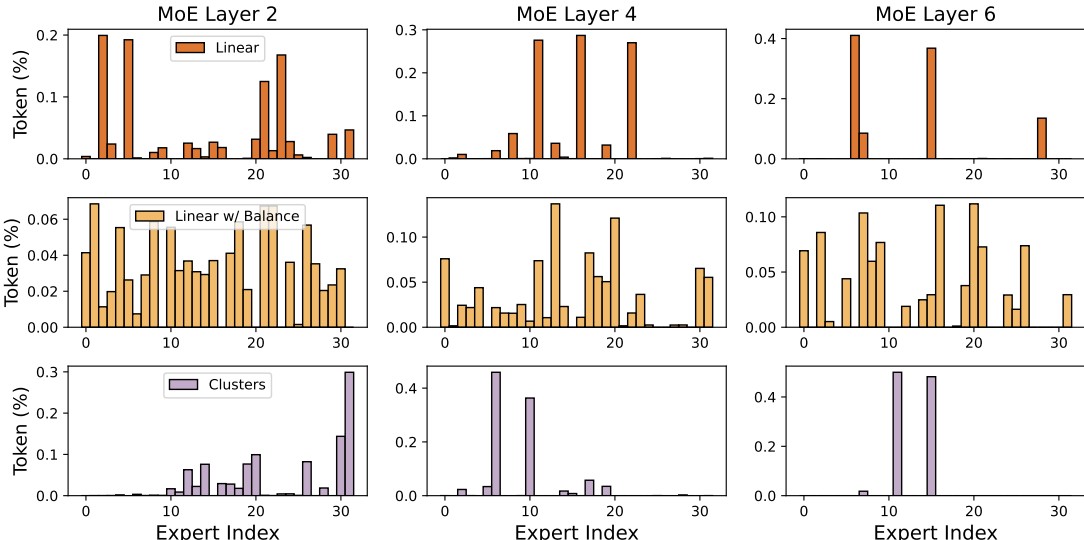

*Figure 9.* Visualization of the distribution of expert allocation for MOIRAI-MOE_S layers 2, 4, and 6 (the last layer) using all data from the Monash benchmark.

token clustering, we present their expert allocation distributions aggregated across all datasets in the Monash benchmark, as illustrated in Figure 9. In terms of selection diversity, we observe the following relationships: Token Clusters (least diverse) < Pure Linear Projection (neutral) < Linear Projection with Load Balancing (most diverse). According to their performance results shown in Figure 4, we can establish the following ranking: Token Clusters > Linear Projection with Load Balancing > Pure Linear Projection. Based on all these observations, we offer the following explanation:

- In the token clusters approach, the expert selections are less diverse because the routing is grounded in pretrained knowledge. The clustering step creates centroids that represent well-structured patterns in the data, and then tokens are routed to specific experts that are particularly suited to handle the type of data represented by their corresponding cluster. While this targeted routing reduces diversity, it enhances performance due to the selection of experts based on more meaningful criteria.

- The addition of load balancing loss increases the diversity of expert selection by spreading the workload and encouraging the use of all experts more evenly. This diversity prevents over-reliance on specific experts, potentially improving generalization and performance compared to pure linear projection. However, this approach might be less targeted than clustering, since it still depends on a learned gating function rather than pretrained centroids.

- In the pure linear projection method, the gating function is entirely learned from scratch. Without any additional constraints (like load balancing), certain experts might get selected more often than others, leading to a neutral level of diversity. Since there is no mechanism to encourage exploration (like load balancing) or specialized routing (like clustering), performance remains lower than the other methods.

### B.6. Visualization of Time Series Observations and Expert Allocations

Following the discussion in the main paper, this section investigates the relationship between raw time series observations and their corresponding expert allocations. In Figure 10, the upper subfigure presents a Traffic Hourly time series sequence with a length of 512. For enhanced visualization, the sequence is segmented using vertical dashed lines, each spanning 16 steps, which is equal to the length of a single time series token. The lower subfigure illustrates the expert allocations at shallow layers for 32 tokens derived from the 512 observations. The yellow straight line represents the specific experts selected by the token at each position. The alignment of subfigures facilitates an intuitive comparison between the time series trends and the associated expert selections.

The figure includes red square boxes to highlight time series segments exhibiting a downward trend followed by a slight upward pattern. These segments consistently correspond to the activation of two specific experts, as shown in the lower

subfigure. This observation suggests that Moirai-MoE effectively captures time-based structures and demonstrates model specialization at the token level.

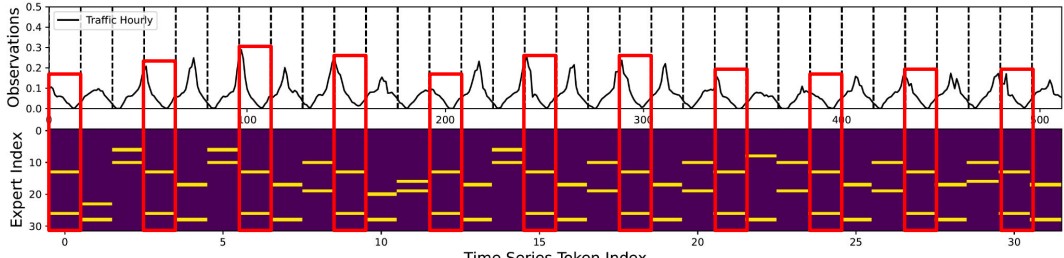

*Figure 10.* Joint visualization of raw time series observations and their corresponding expert selection at shallow layers of MOIRAI-MOE$_\text{S}$. The upper subfigure depicts the raw time series observations with the x-axis representing time step indices (0 to 511). The lower subfigure shows the expert allocation distributions, where the x-axis corresponds to the time series token indices (0 to 31), and the y-axis represents the indices of the 32 experts in the layer.

## C. Limitation

The limitation of this study lies in the efficiency of autoregressive predictions during inference, a well-documented challenge for decoder-only architectures. However, inference solutions developed for large language models (LLMs) could help address this issue. For instance, many LLMs leverage quantization techniques (e.g., 8-bit or 4-bit weights) to significantly reduce computational costs while maintaining performance. In future work, we plan to explore model quantization and pruning methods to optimize efficiency. Additionally, we aim to implement key-value (KV) caching techniques to accelerate inference. However, a key challenge lies in our use of instance normalization, which requires recalculating normalization statistics whenever a new token is generated. This necessity could render the cached hidden states invalid, presenting an obstacle to efficient caching.

## D. Visualization

In this section, we visualize the datasets used in the model analyses (NN5 Daily (Figure 11), Traffic Hourly (Figure 12), and Covid Daily Deaths (Figure 13)) to facilitate understanding of the patterns within the time series data.

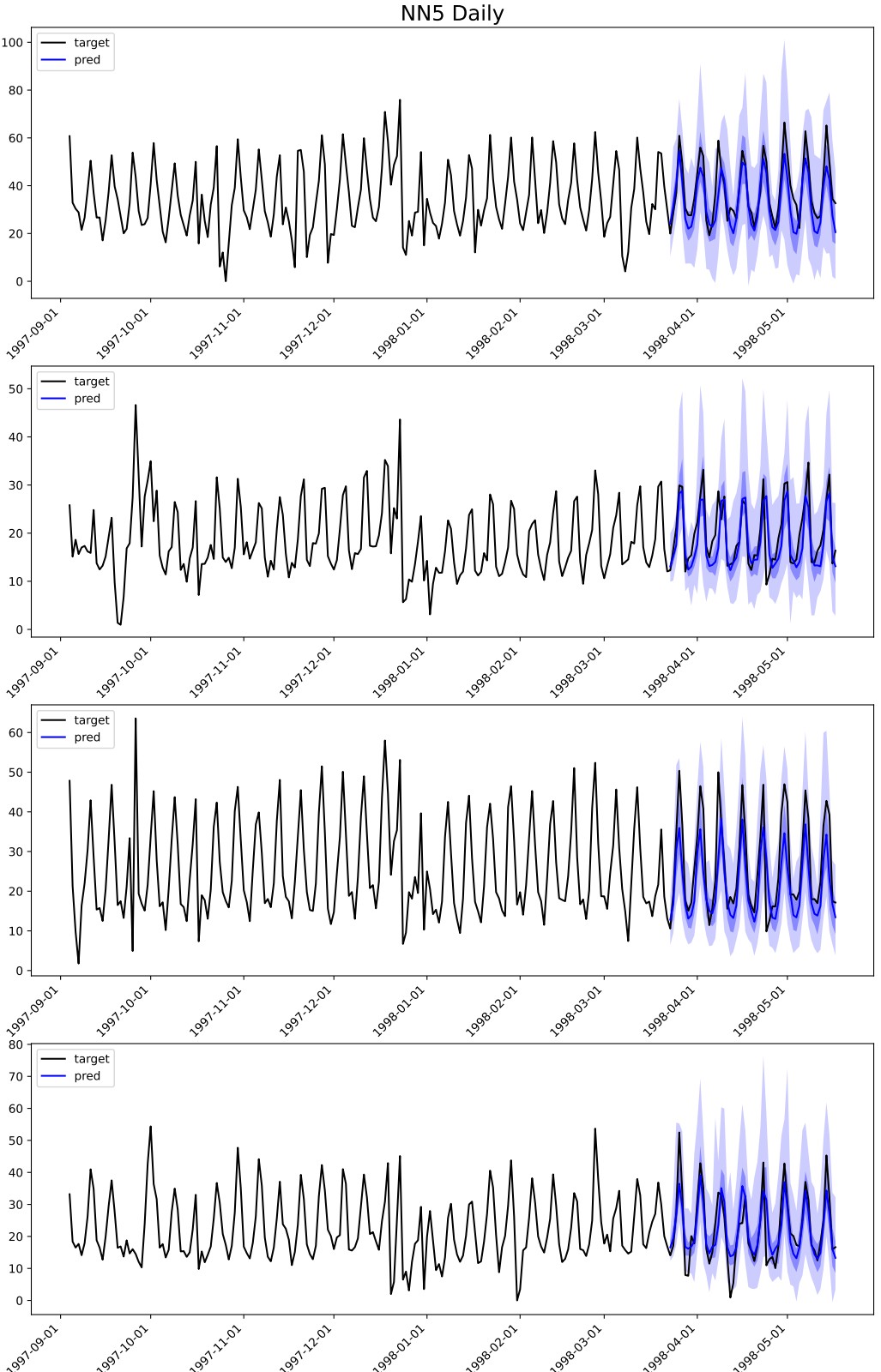

*Figure 11.* Visualization of NN5 Daily data, including both context length and forecast results.

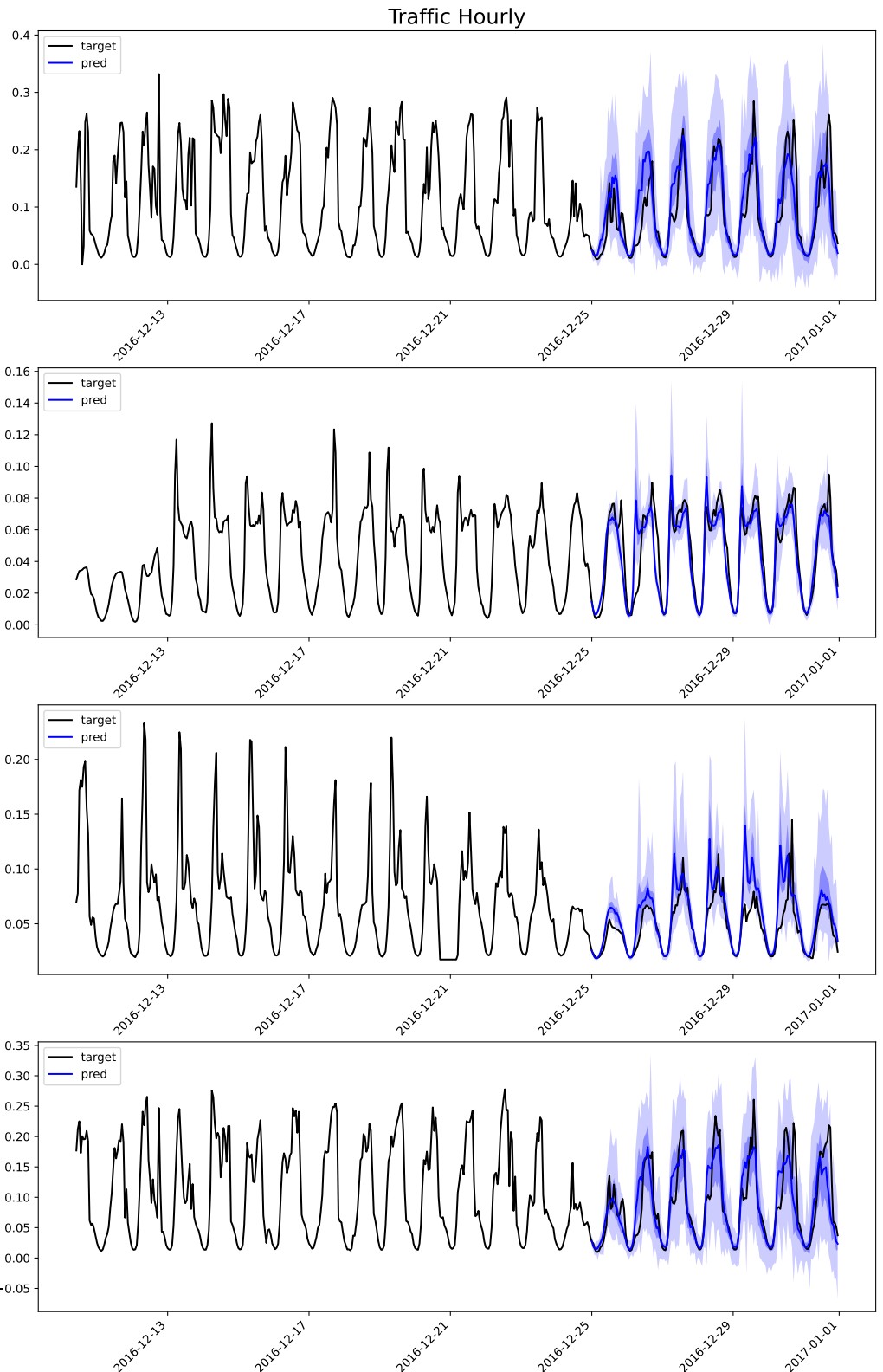

*Figure 12.* Visualization of Traffic Hourly data, including both context length and forecast results.

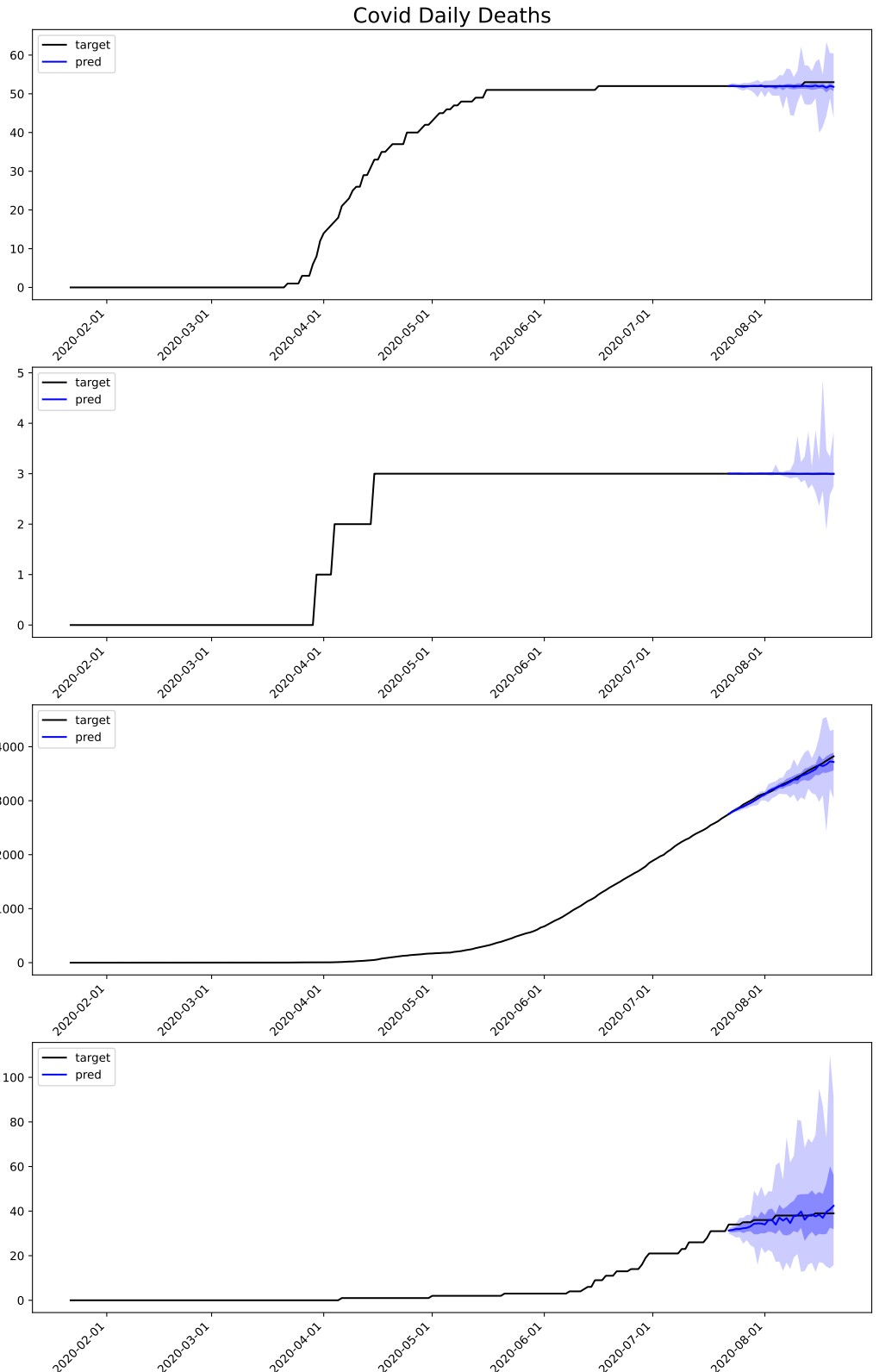

*Figure 13.* Visualization of Covid Daily Deaths, including both context length and forecast results.

