# OpenReview forum: "Moirai-MoE: Empowering Time Series Foundation Models with Sparse Mixture of Experts"
_ICML.cc/2025/Conference — ICML 2025 poster_

### Official Review · Reviewer_Bx6G · 2025-03-09

**Overall Recommendation:** 4

**Summary:**

The authors propose an MoE extension to an existing Time series based foundation model MOIRAI. The authors extends the standard Moirai to MoE to reduce the dependency on human-imposed frequency decomposition as it is not a reliable grouping of pre-training data.  They claim that different frequencies can display similar patterns and similar frequencies can display different patterns. Instead the authors use a sparse MoE approach so that the model can learn the groupings. The authors perform an extensive evaluation and show that their approach results in SOTA performance. The paper is well written and provides clear motivation - this paper is a valuable addition to the scientific community.

## update after rebuttal
After considering the response from the authors - my recommendation remains accept. The work is of merit and useful to the ICML community.

**Claims And Evidence:**

The authors claim that an MoE approach removes the human-imposed frequency decomposition within the MOIRAI model such that the model itself learns the appropriate token clustering. The claims are well founded and the authors provide clear intuition behind their rationale - as seen in Figure 1. Additionally, the authors clearly demonstrate the SOTA performance of their method which further backs up their approach. The claims stated in the paper are well backed up by the evaluation.

**Essential References Not Discussed:**

No

**Experimental Designs Or Analyses:**

The authors use a comprehensive and appropriate experimental design consisting of relevant datasets and other SOTA time-series foundation models.

**Methods And Evaluation Criteria:**

The proposed method is clearly motivated and the evaluation is comprehensive - 39 datasets and current state of the art time-series FMs.
The core idea of MOIRAI-MOE is to exclude human-defined time series groupings while delegating the modeling of diverse time series patterns to the sparsely activated experts in Transformer layers. The authors also investigate existing expert gating functions that generally use a randomly initialized linear layer for expert assignments. They introduce a new expert gating function for accurate expert assignments and improved performance. The author propose automatic time series token-level specialisation, where diverse tokens are handled by different experts, while similar tokens share parameter space, reducing learning complexity. MoE is done by replacing every FFN by an MoE layer consisting of M experts and a Gating function G. The evaluation criteria used is well motivated and comprehensive.

**Other Comments Or Suggestions:**

One typo on line 213: "Mo" should be "MoE"

**Other Strengths And Weaknesses:**

A weakness of this study is around the evaluation of the probabilistic forecasting component of the paper - which I think is more valuable than the deterministic evaluations performed. The results are indeed impressive but a statistical analysis of the probabilistic forecasting performance and characteristics would be a valuable addition to this well motivated study.

**Questions For Authors:**

1) The authors argue that different frequencies can display similar patterns and similar frequencies can display different patterns - which I agree with. However could some wavelet transformation/layer not alleviate this issue within the standard architecture?

2) No adaptive patching was considered in the method - so how are different resolutions considered if at all?

3) The authors perform mini-batch k-means clustering on the attention outputs to continuously update the cluster centroids at each layer where the cluster number is equal to the number of experts - did you observe any discrepancies doing this such as the number of clusters being significantly different to the number of experts?

**Relation To Broader Scientific Literature:**

This paper is of clear interest to time-series forecasting and advancing the current SOTA with respect to time-series Foundation models. The impact of such research is of high significance and impact to the wider scientific community due to the application of time-series forecasting.

**Theoretical Claims:**

No theoretical claims were made in the paper

---

> ### Author Rebuttal · Authors · 2025-04-01
>
> **[C1] A weakness of this study is around the evaluation of the probabilistic forecasting component of the paper - which I think is more valuable than the deterministic evaluations performed. The results are indeed impressive but a statistical analysis of the probabilistic forecasting performance and characteristics would be a valuable addition to this well motivated study.**
>
> We provide standard deviations and statistical significance in Table 1 at this link: https://drive.google.com/file/d/1bwJ7dyji_OnSNkYXpA6IOpwnvF6nOZmS/view
>
> **[C2] The authors argue that different frequencies can display similar patterns and similar frequencies can display different patterns - which I agree with. However could some wavelet transformation/layer alleviate this issue within the standard architecture?**
>
> Thank you for the insightful comment. Indeed, applying a wavelet transformation in the input space can effectively help identify and distinguish patterns in time series. However, pattern specialization of Moirai-MoE are performed at each Transformer layer, and this allows our model to adaptively capture patterns at various hierarchical abstraction levels throughout the model. This hierarchical and adaptive specialization provides greater flexibility and modeling power compared to a static transformation applied at the input. Nevertheless, your suggestion of integrating wavelet transformations into the input is intriguing, and further experimentation would be valuable to empirically evaluate and compare the performance.
>
> **[C3] No adaptive patching was considered in the method - so how are different resolutions considered if at all?**
>
> Thank you for your comment. We would like to clarify that different resolutions are considered equivalent in our context, as we use a uniform patch size. The critical factor in our approach is the pattern or shape of the time series data, and the specialization of the model primarily addresses this aspect of diversity. Additionally, we have already included experiments examining the effects of different patch sizes in the appendix. The results show that the choice of patch size 16 works the best.
>
>
> **[C4] The authors perform mini-batch k-means clustering on the attention outputs to continuously update the cluster centroids at each layer where the cluster number is equal to the number of experts - did you observe any discrepancies doing this such as the number of clusters being significantly different to the number of experts?**
>
> Thank you for the insightful question. This question depends on the relative values of the cluster number K and the number of experts E, and we address it in two parts:
>
> Case 1: K smaller than E — This setting is not valid in our framework. Since each cluster corresponds to an expert during MoE training, having fewer clusters than experts would result in multiple experts sharing the same cluster centroid. This leads to repeated expert selections and undermines the intended diversity and specialization of the experts.
>
> Case 2: K larger than E — This is a more interesting scenario. For example, when K=64 and E=32, we face the challenge of mapping the 64 cluster centroids to the 32 experts. A practical and effective strategy we adopt is as follows: for each token (data point), we compute its distance to all 64 centroids, and select the top 32 centroids with the smallest average distances across tokens. These selected centroids are then used during MoE training. Despite having more clusters than experts, this approach still allows us to select the most representative centroids for expert routing.
>
> We validated this approach experimentally. Results show that this method achieves comparable performance to the standard setting where K equals to E, indicating that even with a larger number of clusters, a careful selection mechanism can yield effective and robust MoE behavior.

---

### Official Review · Reviewer_JU2u · 2025-03-12

**Overall Recommendation:** 2

**Summary:**

This paper proposes MOIRAI-MOE, a time series foundation model based on the Mixture of Experts (MoE). It replaces the traditional frequency-grouping strategy with data-driven token-level specialization, thus addressing the pre-training challenges posed by the high heterogeneity of time series data. The model achieves significant performance improvements on 39 datasets by introducing an expert routing mechanism based on the clustering centers of pre-trained dense models and an autoregressive prediction objective.

**Claims And Evidence:**

Claims are supported by in-distribution/zero-shot results in Tables 3, 6, and 8, which show MOIRAI-MOE’s superiority in MAE, CRPS, and other metrics. Ablation studies demonstrate that MoE specialization, not just the decoder objective, drives performance gains.

**Essential References Not Discussed:**

No critical omissions noted. The paper comprehensively cites relevant works like TimesFM, Chronos, and existing MoE literature.

**Experimental Designs Or Analyses:**

This paper includes baseline comparisons of both foundation models (MOIRAI, Chronos) and specialized models (PatchTST). Meanwhile, ablation studies isolate the impact of MoE, gating, and pretraining objectives. Moreover, the efficiency analysis in Table 4 shows comparable inference speeds to MOIRAI despite larger total parameters.

**Methods And Evaluation Criteria:**

The sparse MoE efficiently scales model capacity while maintaining computational efficiency.  Cluster Gating improves load balancing and expert relevance compared to random initialization.
The evaluation uses diverse datasets (Monash, zero-shot benchmarks) and metrics (MASE, CRPS), ensuring robustness.

**Other Comments Or Suggestions:**

N/A

## update after rebuttal
While the authors have addressed certain technical concerns (e.g., pruning validation and training cost metrics) with additional clarifications, some limitations around generalizability dependencies and methodological novelty remain unresolved. Though the reviewer leans toward maintaining the original score, the paper's demonstrated empirical improvements and structured rebuttal make a case for acceptance, depending on broader reviewer consensus.

**Other Strengths And Weaknesses:**

Strengths:
1. This paper proposed a new model architecture, which introduces a Sparse Mixture of Experts (MoE) to dynamically route tokens to specialized experts, overcoming the limitations of human-defined frequency grouping.
2. The proposed method proposes a novel gating mechanism using cluster centroids from pre-trained dense models, outperforming randomly initialized linear routing.
3. The proposed method achieves superior performance, which outperforms state-of-the-art baselines (e.g., MOIRAI, Chronos) on 39 datasets, reducing MAE by up to 17% with the same activated parameters and surpassing others with 65× fewer activated parameters.

Weaknesses:
1. Analysis in Fig. 6 reveals some experts in deeper layers are rarely activated, indicating inefficient parameter usage. While pruning is mentioned as future work, no concrete solution is provided.
2. The proposed clustering-based routing relies heavily on pre-trained MOIRAI, which introduces a dependency on the quality of the pre-trained model and dataset (LOTSA), and thus potentially limiting generalization to domains outside the pretraining distribution.
3. MOIRAI-MOE contains significantly more parameters than those of MOIRAI. This increases memory demands during training and deployment. Is it possible to use model compression technology to reduce the parameters of the model?
4. When MOIRAI-MOE and MOIRAI parameters are equivalent, how does MOIRAI-MOE perform?
5. While outperforming foundation models, MOIRAI-MOE still lags behind fully fine-tuned specialized models like TiDE on some datasets (e.g., ETT1 in Table 8), indicating room for improvement in extreme domain adaptation.

**Questions For Authors:**

See weaknesses.

**Relation To Broader Scientific Literature:**

Builds on MOIRAI (Woo et al., 2024) by replacing frequency projections with MoE. Extends sparse MoE from language/vision (Fedus et al., 2022) to time series. Compares favorably with concurrent work Time-MoE (Shi et al., 2024) by introducing cluster gating and patch tokenization.

**Theoretical Claims:**

Theoretical derivations (e.g., MoE layer formulation, gating function) are standard and correctly applied. However, the theoretical grounding (e.g., convergence guarantees) of the cluster-based gating mechanism is not explicitly proven.

---

> ### Author Rebuttal · Authors · 2025-04-01
>
> **[C1] While pruning is mentioned as future work, no concrete solution is provided.**
>
> Thanks for the comment. A concrete pruning solution is to first evaluate expert usage by tracking gating activations during pretraining. Identify experts with significantly fewer activations (e.g., activated less than 1\% of total gating decisions) as underutilized. Completely remove these expert modules from the model architecture, thereby reducing GPU memory usage. Lastly, update the gating network accordingly to ensure it routes inputs correctly to the remaining experts.
>
> **[C2] The proposed clustering-based routing relies heavily on pre-trained MOIRAI, which introduces a dependency on the quality of the pre-trained model and dataset (LOTSA), and thus potentially limiting generalization to domains outside the pretraining distribution.**
>
> Regarding the argument about "potentially limiting generalization to domains outside the pretraining distribution": This challenge is inherent to all time series foundation models, not solely Moirai-MoE. Therefore, it underscores the importance of assembling a comprehensive and diverse time series pretraining corpus to maximize coverage of potential application domains.
>
> Regarding the argument on the "dependency on the quality of the pretrained model": We acknowledge this dependency. Indeed, recent advancements in MoE-based Large Language Models (such as Qwen1.5-MoE) explicitly leverage pretrained dense models to introduce beneficial inductive biases, thereby enhancing model performance. Consequently, we consider this dependency a strategic strength rather than a limitation. Our work aligns with this research direction, with the assumption of a high-quality pretrained model and focusing on effectively leveraging it to achieve further performance gains.
>
> **[C3] MOIRAI-MOE contains significantly more parameters than those of MOIRAI. This increases memory demands during training and deployment. Is it possible to use model compression technology to reduce the parameters of the model?**
>
> The increase in parameter count of Moirai-MoE does not proportionally affect computational cost during inference because, at any given time, only a subset of the experts are activated. However, model compression techniques can indeed be leveraged to reduce the parameters: (1) By converting weights from full precision (e.g., FP32) to lower precision (e.g., INT8 or FP16), quantization techniques can substantially reduce memory footprint while maintaining accuracy. (2) Techniques such as knowledge distillation can transfer learned representations from a larger Moirai-MoE model to a smaller, compressed model, thus achieving efficiency without a significant loss in performance.
>
> Implementing these compression techniques individually or in combination could mitigate memory concerns associated with Moirai-MoE. We will explore integrating these strategies into our framework and evaluate their impact on model performance and efficiency in subsequent research. Thank you for the suggestion.
>
> **[C4] When MOIRAI-MOE and MOIRAI parameters are equivalent, how does MOIRAI-MOE perform?**
>
> Our current configuration is as follows: Moirai-Base has a total of 91M parameters, while Moirai-MoE-Small has 117M parameters. To address your concern, we introduce a variant of Moirai-MoE-Small, called Moirai-MoE-Small-V2, by reducing the FFN dimension. This variant has a total parameter count of 89M, which is closely aligned with that of Moirai-Base.
>
> We pretrain Moirai-MoE-Small-V2, evaluate it on the 29 datasets of the Monash benchmark, and compute the aggregated performance. Note that the aggregated MAE of Moirai-Base is 0.71 (according to Figure 3 in the main paper), while Moirai-MoE-Small-V2 achieves an aggregated MAE of 0.67, demonstrating superior performance.
>
> **[C5] MOIRAI-MOE still lags behind fully fine-tuned specialized models like TiDE on some datasets (e.g., ETT1 in Table 8), indicating room for improvement in extreme domain adaptation.**
>
> We believe there might be some misunderstanding. We carefully compare the performance of TiDE and Moirai-MoE on the ETT1 dataset in Table 8. The CRPS and MASE of TiDE are 1.056 and 6.898, while the CRPS and MASE of Moirai-MoE-Small are 0.288 and 1.750. Since lower values are better, Moirai-MoE-Small is actually significantly better than TiDE on this dataset.
>
> However, to address your concern, we do find that on the Walmart dataset, TiDE performs better than Moirai-MoE-Small: TiDE scores 0.077 for CRPS and 0.814 for MASE, while Moirai-MoE-Small scores 0.090 for CRPS and 0.927 for MASE. This is a case where Moirai-MoE requires fine-tuning for better domain adaptation. After fine-tuning Moirai-MoE-Small, the results improve to 0.071 for CRPS and 0.756 for MASE, successfully surpassing TiDE.

---

> > ### Comment · Reviewer_JU2u · 2025-04-08
> >
> > Thank you for the authors' response. However, I find that my key concerns remain largely unaddressed and I will maintain my original score, detailed in the following:
> >
> > **Re. for C1:**
> >  Although the authors discussed a potentially viable pruning method, which removed the underutilized branches. However, it is only a very common and simplistic pruning paradigm. The authors did not consider how pruning specifically applies to time series models, nor how to mitigate the severe performance degradation that may result from pruning.
> >
> > **Re. for C2:**
> >  I acknowledge that “this challenge is inherent to all time series foundation models,” and indeed, it is an important issue that needs to be addressed. The proposed method, which is based on pre-trained MOIRAI, does not consider how to tackle this challenge, thus limiting the overall effectiveness of the approach.
> >
> > **Re. for C3:**
> >  The authors claim that “the increase in parameter count of Moirai-MoE does not proportionally affect computational cost during inference.” However, the computational and memory costs during training are also crucial. In general, the authors did not provide concrete measurements of computational or storage cost, either during training or inference. Therefore, their explanation is not sufficiently convincing.
> >
> > **Re. for C4:**
> >  I noticed that MOIRAI-MoE-Small (117M parameters) has significantly more parameters than MOIRAI-Small (only 14M), indicating that the MoE mechanism introduces a substantial increase in parameter count. The authors compared MOIRAI-MoE-Base with MOIRAI-MoE-Small, which is an unfair comparison. Furthermore, whether the MoE architecture introduces additional computational overhead should be discussed in more detail.
> >
> > Therefore, I will keep the original rating.

---

> > > ### Author Response · Authors · 2025-04-08
> > >
> > > Dear reviewer, we would like to express our sincere thanks for the time you have taken to review our submission, and thank you very much for responding our rebuttal. Please find our further responses below.
> > >
> > > **[C1] Although the authors discussed a potentially viable pruning method, which removed the underutilized branches. However, it is only a very common and simplistic pruning paradigm. The authors did not consider how pruning specifically applies to time series models, nor how to mitigate the severe performance degradation that may result from pruning.**
> > >
> > > Thank you very much for raising this point. After our initial response four days ago, we implemented the discussed pruning method, pretrained the resulting model, and evaluated its performance on the 29 datasets from the Monash benchmark. So the aggregated results are now available and they show that the pruned Moirai-MoE-Small achieves an aggregated MAE of 0.66, which is comparable to the original Moirai-MoE-Small’s aggregated MAE of 0.65. These outcomes are reasonable, as underutilized experts do not contribute to the model's capabilities, meaning their removal does not negatively impact overall performance.
> > >
> > >
> > > **[C2] I acknowledge that “this challenge is inherent to all time series foundation models,” and indeed, it is an important issue that needs to be addressed. The proposed method, which is based on pre-trained MOIRAI, does not consider how to tackle this challenge, thus limiting the overall effectiveness of the approach.**
> > >
> > > Thank you for your comment. The overall effectiveness of Moirai-MoE has been thoroughly validated through comprehensive evaluations on 39 datasets, outperforming state-of-the-art baselines (e.g., Moirai, Chronos, TimesFM). The challenge mentioned in our current discussion pertains primarily to the comprehensiveness of our pretraining dataset, LOTSA. This issue relates more directly to the coverage of the pretraining corpus rather than to methodological considerations, placing it somewhat outside the intended scope of our current paper. Our work focuses explicitly on methodological advancements, specifically enhancing the Moirai model architecture through our proposed Moirai-MoE architecture. Therefore, we kept the pretraining corpus fixed as a controlled factor to demonstrate and discuss improvements attributed solely to the methodological innovation.
> > >
> > >
> > > **[C3] The computational and memory costs during training are also crucial. In general, the authors did not provide concrete measurements of computational or storage cost, either during training or inference.**
> > >
> > > Thank you for raising this point. We provided inference cost details in Table 4 of our paper, but we acknowledge that the training cost details were not fully addressed. To clarify the computational and memory costs during training, we inspect the pretraining logs and provide the following information:
> > >
> > > We utilized 16 GPUs to pretrain both Moirai and Moirai-MoE models. Specifically: (1) Moirai-MoE-Small required 9.49 hours of pretraining for 50,000 steps, with a peak memory usage of 14.52 GB per GPU. (2) Moirai-Small took 4.69 hours to pretrain for 50,000 steps, with a peak memory usage of 11.45 GB per GPU. The difference in pre-training time and memory usage is due to the difference in the total number of parameters between Moirai-Small (14M) and Moirai-MoE-Small (117M).
> > >
> > >
> > > **[C4] The authors compared MOIRAI-MoE-Base with MOIRAI-MoE-Small, which is an unfair comparison. Furthermore, whether the MoE architecture introduces additional computational overhead should be discussed in more detail.**
> > >
> > > Thank you for your comment. In our initial rebuttal, we explicitly stated that our comparison is between Moirai-Base (91M) and Moirai-MoE-Small-V2 (89M), which is fair, given their similar parameter counts. We did not compare Moirai-MoE-Base and Moirai-MoE-Small in our initial response. Your concern about the computational cost associated with the MoE architecture is outlined above in the response to [C3].
> > >
> > > Thank you again for your constructive feedback. We hope our additional clarifications address the concerns and would greatly appreciate if you could reconsider the rating.

---

### Official Review · Reviewer_Bkoz · 2025-03-12

**Overall Recommendation:** 3

**Summary:**

This paper introduces a novel foundational model for time series forecasting, building upon the architecture of Moirai. The primary motivation is to address a key limitation of existing time series foundational models, which rely on manually imposed clustering—such as specialized layers for different time series structures (e.g., frequency-based clustering in Moirai). Instead, the authors propose an automated approach using a mixture of experts (MoE) to dynamically detect dataset diversity and automatically cluster heterogeneous patterns within the pretraining data. Additionally, rather than using conventional MoE layers, which are sensitive to initialization, the authors introduce a clustering-based method that enhances robustness. Experimental results on multiple datasets demonstrate the effectiveness of the proposed approach compared to existing methods.

**Claims And Evidence:**

Key claims of the authors with comments

**Improved Token-Level Clustering**
Moirai-MoE enables automatic and more efficient token-level clustering compared to Moirai, which relies on human-defined frequency-based clustering. Figure 5 illustrates how Moirai-MoE performs data-driven clustering. However, there is no empirical evidence showing that this clustering is interpretable by humans. The appendix should include more visualizations, highlighting cases where clustering does or does not align with human intuition. Additionally, testing on synthetic datasets with clear clustering structures could help formally validate this approach. For example, experiments could assess clustering uniqueness in homogeneous datasets or evaluate whether Moirai-MoE correctly identifies imposed structures in datasets with predefined clusters. Stronger empirical evidence is needed.

**Improved Expert Gating in MoE**
Moirai-MoE enhances MoE by introducing a clustering-based expert gating mechanism, addressing the sensitivity issue in traditional MoE architectures. Figure 4 presents an ablation study comparing this gating method to standard linear projection. While the experiments support this claim, further validation is needed. Running multiple trajectories with different initializations could help assess the stability of clustering beyond performance improvements. Additionally, reporting standard deviations would strengthen the claim regarding Moirai-MoE’s robustness.

**State-of-the-Art (SOTA) Performance Across Benchmarks**
The authors conduct extensive benchmarking against multiple competitors and datasets. However, incorporating statistical tests, such as reporting standard deviations and statistical significance, would better demonstrate the stability and reliability of the proposed approach.

**Essential References Not Discussed:**

No essential references to mention

**Experimental Designs Or Analyses:**

**Experimental Results & Reproducibility**

The experimental results demonstrate computational efficiency and good performance, but several improvements might improve the manuscript.

A. Statistical Validation

- As previously mentioned, incorporate standard deviations where necessary.
- Perform statistical tests to confirm the stability of the approach.

B. Reproducibility & Transparency

- Clearly explain how results were collected (e.g., best epochs for validation/test/train, last epoch for in-distribution experiments).
- Provide a detailed explanation of how "out-of-distribution" data is defined and ensure no data leakage occurs.
- Include algorithmic details (Mainly in algorithm environment) in the manuscript to facilitate reproducibility.
C. Visualization of Zero-Shot Forecasting

- Include additional visualizations (at least in the appendix) showcasing zero-shot forecasting results even though some are given but not compared with competitors.
- Specifically, highlight both successes and failures, explaining where Moirai fails and why, as well as where it succeeds (or Moirai-MoE) and the underlying reasons.

**Methods And Evaluation Criteria:**

**Relevant Benchmark Datasets**
The authors primarily use the Monash dataset, which is increasingly recognized in the time series literature, making it a sensible choice for benchmarking.

**Valid Methodological Approach**
The proposed methods are well-founded, as they address data heterogeneity—a key challenge in time series foundational models.

**Other Comments Or Suggestions:**

All suggestions have been written before

**Other Strengths And Weaknesses:**

**Strengths**

- Addressing data heterogeneity is a valuable approach, and the use of MoE appears to be a promising solution.
- The paper includes extensive experiments across multiple datasets, demonstrating thorough evaluation.
- The writing is clear, making the paper easy to follow.

**Weaknesses**

- The contributions are somewhat moderate, as several prior works have already explored MoE for time series (e.g., Time-MoE).
- There is a lack of strong theoretical justification and limited empirical evidence regarding the stability of the expert gating system.

**Questions For Authors:**

**On the expert gating**

Instead of using a direct clustering approach for expert gating, introducing a third loss (a clustering loss) in the embedding space could be beneficial. This would encourage natural grouping of similar patterns without requiring an explicit, manually determined clustering step. A contrastive loss (e.g., a soft clustering loss or DeepCluster-like approach) could be used to enforce structure in the learned representation. This avoids the need for a preliminary run of Moirai with a single layer, making the process more automated and adaptive. Have you considered loss functions like InfoNCE or DeepCluster for this purpose?

**Computational Comparison with Moirai**

- Training Complexity:
Moirai already reduces the sample size by clustering, but if clustering is done implicitly through a loss function rather than manually, the computational overhead may shift from pre-processing to training.
If the gating mechanism remains data-dependent, backpropagation complexity might increase due to additional constraints in the loss function.
- Inference Complexity:
If expert selection remains soft, inference might require evaluating multiple experts per sample, increasing computation time.

Could you comment about time complexity and memory complexity of the algorithm and illustrate that empirically.

**Relation To Broader Scientific Literature:**

**MoE in Time Series Foundational Models** – *Weak Contribution*
The incorporation of MoE is not a significant contribution, as it has already been used in Time-MoE. While the authors highlight some differences, the novelty remains limited.

**New Expert Gating with Clustering** – *Moderate Contribution*
This contribution is moderately novel, as such expert gating has not been previously proposed. However, the lack of theoretical evidence and extensive experimental validation weakens the claim of its stability.

**Theoretical Claims:**

The manuscript does not provide a theoretical guarantee for the proposed method. To enhance its quality, the authors might conduct a theoretical analysis comparing clustering-based MoE to linear mapping. Even in a simplified setting—where each expert is linear—they could analyze how a student MoE approximates a teacher MoE under both architectures (for examples). This would provide deeper insights into the model’s theoretical foundations. But this is more a suggestion than a criticize.

---

> ### Author Rebuttal · Authors · 2025-04-01
>
> **IMPORTANT: Figures 1, 2, 3 and Tables 1, 2 are provided here: https://drive.google.com/file/d/1bwJ7dyji_OnSNkYXpA6IOpwnvF6nOZmS/view**
>
> **[C1] Testing on synthetic datasets with clear clustering structures. Assess clustering uniqueness in homogeneous datasets or evaluate whether Moirai-MoE correctly identifies structures in datasets with predefined clusters.**
>
> We generate synthetic datasets with basic time series patterns to assess the ability of Moirai-MoE in a controllable way. The resulted visualization is in Figure 1.
>
> **[C2] Include zero-shot visualizations, highlight both successes and failures, explaining where Moirai fails and why**
>
> We conducted additional visualizations comparing Moirai-MoE-Small with Chronos-Small and TimesFM, as shown in Figures 2 and 3. Figure 2 illustrates failure cases for Moirai-MoE—these time series primarily exhibit trend without seasonality. In such cases, both Moirai-MoE-Small and Chronos-Small perform poorly, while TimesFM performs exceptionally well. However, we suspect possible data leakage in TimesFM, as its forecasts align almost perfectly with the future trend, which we believe could be inherently unpredictable. Figure 3 presents success cases where Moirai-MoE outperforms the others. Its forecasts are generally closer to the ground truth, whereas Chronos-Small and TimesFM tend to show larger overestimations or underestimations. Due to word count limitations at the rebuttal stage, we present only two cases here, but we will add more cases and analyses in the appendix of the manuscript.
>
> **[C3] Reporting standard deviations and statistical significance would better demonstrate the stability and reliability of the proposed approach**
>
> Standard deviations and statistical significance are in Tables 1 and 2.
>
> **[C4] Clearly explain how results were collected (e.g., best epochs for validation/test/train, last epoch for in-distribution experiments)**
>
> Moirai-MoE is pretrained for a certain number of epochs on a large time series corpus. After pretraining, the checkpoint saved from the last epoch is used to perform inference on 29 in-distribution datasets. Based on the aggregated performance, we know the optimal number of epochs required to achieve the best in-distribution performance and then we use this checkpoint for zero-shot evaluation.
>
> **[C5] Provide explanation of how "out-of-distribution" data is defined and ensure no data leakage occurs.**
>
> We follow the settings of the well-recognized time series foundation models (TSFMs), such as Moirai, Chronos, and TimesFM, the out-of-distribution data refers to those datasets (including their training, validation, and test sets) that are not included in the pretraining corpus.
>
> **[C6] The contributions are somewhat moderate. The incorporation of MoE is not a significant contribution, as it has already been used in Time-MoE.**
>
> We would like to argue our position from the following points. First, although Time-MoE has applied MoE to TSFMs, its application has not demonstrated superior effectiveness, as evidenced by its inferior zero-shot forecasting performance. In our zero-shot evaluations, its performance significantly trails behind Moirai-MoE. It does not even surpass Moirai-Large, as demonstrated in both our zero-shot evaluations and the point forecasting results over 27 datasets reported by the Chronos Team (https://aws.amazon.com/blogs/machine-learning/fast-and-accurate-zero-shot-forecasting-with-chronos-bolt-and-autogluon/). Hence, our contribution lies not merely in applying MoE to TSFMs but, importantly, in achieving notably superior performance compared to Time-MoE. Second, we have dedicated considerable effort to exploring and understanding the internal mechanisms of MoE models. This depth of analysis has not been previously addressed in the literature, including the Time-MoE study. Thus, this should be recognized as a meaningful contribution of our paper.
>
> **[C7] Have you considered InfoNCE or DeepCluster for this purpose? Could you comment about time complexity and memory complexity.**
>
> We appreciate the suggestion regarding the integration of clustering-based losses, such as InfoNCE or DeepCluster, into MoE training. We also find your provided training and inference complexity analysis reasonable. While these methods could indeed offer a more automated approach to structuring representations, their effectiveness might not be guaranteed in practice. Imposing clustering losses during MoE training could introduce optimization challenges and potentially interfere with the primary learning objectives of the model. Our current approach adopts a two-stage process. This separation offers training stability, particularly in the early stages when representations may not be sufficiently structured for effective contrastive learning. Our method can also leverage the inductive bias from the pretrained Moirai model. Nevertheless, we consider your recommendation a valuable and promising direction for future exploration.

---

> > ### Comment · Reviewer_Bkoz · 2025-04-04
> >
> > I would like to thank the author for their detailed comments and thorough evaluation. Empirically, the results are quite interesting, even though the performance gap compared to competitors like TimesFM and Chronos remains somewhat limited. Nonetheless, the empirical efforts are valuable, and the results could be of interest to the community, which is why I have decided to maintain my score as "weak accept."
> >
> > However, I find the methodology to be somewhat empirical, with limited theoretical justification and strong empirical evidence, aside from the motivation for automatic clustering. Additionally, the contribution feels moderate, with the exception of the promising results that are claimed.
> >
> > From my perspective, the key strengths of the method lie in the good empirical results across different datasets and the motivation behind automatic clustering. However, I feel that the theoretical investigation and the underlying intuitions are not fully developed, which limits the potential for broader impact within the community together with the limited contribution which is a bit incremental.
> >
> > For these reasons, I am keeping my score unchanged.

---

### Official Review · Reviewer_h3gH · 2025-03-15

**Overall Recommendation:** 2

**Summary:**

The paper focuses on the pretraining of time series foundation models using large time series corpora.
The paper argues that there are significant drawbacks to current approaches that address heterogeneity by grouping time-series based on human-identified features such a frequency. The paper proposes an alternative that involves using a sparse mixture of experts (MoE) that provides an avenue towards automatic specialization. The paper reports results for multiple datasets, demonstrating outperformance of the selected baselines. Experimental analysis is included that explores the operation of the MoE foundation models.

--AFTER REBUTTAL
I thank the authors for their thoughtful response. I was somewhat surprised that there was no response to my follow-up questions about the routing mechanism and the very sparse utilization of experts. The authors addressed three of my initial concerns, providing good responses. In particular, they emphasized the value of the engineering effort in getting an MoE time-series foundation model to work and they reported variability for some experiments. The authors pointed out that I had misinterpreted the gating mechanism and that it was non-trainable. But understanding this, the follow-up question in my response to the rebuttal was how the non-trainable router differed from a clustering-based hash approach that has been proposed previously. There was no response to this. The authors were claiming this as a significant novel technical contribution of the paper, so the lack of justification is concerning. The other residual concern was how uneven the usage of the experts was. I had concerns about the design approach given that it emerges that only a handful of the experts are being used at higher layers in the architecture. Given that these concerns were not addressed, I will retain my score (weak reject).

**Claims And Evidence:**

Please see "Other Strengths and Weaknesses"

**Essential References Not Discussed:**

Please see "Other Strengths and Weaknesses"

**Experimental Designs Or Analyses:**

Please see "Other Strengths and Weaknesses"

**Methods And Evaluation Criteria:**

Please see "Other Strengths and Weaknesses"

**Other Comments Or Suggestions:**

None

**Other Strengths And Weaknesses:**

Strengths

S1. The paper proposes to augment a foundational time series model (MOIRAI) with a sparse Mixture-of-experts. The key innovation is a novel strategy to pre-train the experts, which involves clustering tokens according to the attention outputs of an inference model. There is also a carefully executed tokenization strategy.

S2. The paper reports the results of extensive experiments. These demonstrate that the MoE approach achieves a significant performance improvement over the base MOIRAI models, even when there are far fewer active parameters.

S3. The paper conducts a good experimental analysis to provide insights into how the MoE is behaving (e.g., how different frequencies are assigned to different experts at different layers of the model). There are useful visualizations and thorough investigations in the appendices. The paper includes ablation studies as well as studies of the sensitivity to different design choices and parameter settings.

Weaknesses

W1. The technical contributions of the paper are relatively minor. There is some adjustment of the token construction process, but this appears to be more of an engineering exercise to make the operation and training efficient. Beyond this, the primary novel technical contribution is the use of the token clusters during the pre-training of the MoE, with the clustering based on the attention outputs of an inference model. While this is an innovative solution, and a welcome acknowledgement of the challenge of successfully training a sparse MoE, the technical content is presented in 13 lines of text and a single equation. Overall, while there are some innovative contributions, the paper seems to be a relatively straightforward application of existing sparse MoE techniques to MOIRA, with some careful engineering effort to ensure that the training is successful.

W2. The paper claims to introduce “a new expert gating function for accurate expert assignments and improved performance”. This claim does not seem to be supported, as I cannot detect a significant difference between the proposed gating and the procedure in Shazeer et al, 2017. There is novelty in the pretraining based on clusters but this does not imply a new gating function – the claim is misleading. A more correct claim would be “we introduce a novel procedure for pre-training the gating function in a standard sparse mixture-of-experts model”.

W3. The paper does not report variability for any of the experiments and there are no confidence intervals. There are no statistical significance tests.

**Questions For Authors:**

Q1.	Figures 5 and 6 (and 9 in the appendix) show why expert allocation is important. This paper uses a standard Sparse-MoE setup with all experts having the same structure. If trained and regularised correctly, allocation should be approximately uniform (like Fig. 7 in Mixtral). While it is desirable to have some experts focused on specific frequencies, Figure 6 suggests that many experts are not contributing at all. This is also confirmed by Figure 9 in the Appendix. Although the token clusters approach achieves the best performance, these results suggest that the training is not operating correctly. It suggests some form of representation collapse. Was this investigated by the authors?


[R1] Jiang, Albert Q., et al. "Mixtral of experts." arXiv preprint arXiv:2401.04088 (2024).

**Relation To Broader Scientific Literature:**

Please see "Other Strengths and Weaknesses"

**Theoretical Claims:**

Please see "Other Strengths and Weaknesses"

---

> ### Author Rebuttal · Authors · 2025-04-01
>
> **[C1] Code repository are corrupted.**
>
> The issue appears to be related to viewing the files directly through the web interface. We verify that downloading the repository locally resolves the issue.
>
> **[C2] The paper seems to be a relatively straightforward application of existing sparse MoE techniques to MOIRAI, with some careful engineering effort to ensure that the training is successful.**
>
> Regarding the argument "the token construction process is more of an engineering exercise to make the operation and training efficient; with some careful engineering effort to ensure that the training is successful". In fact, efficiency is a crucial aspect, particularly when developing Time Series Foundation Models (TSFMs). TSFMs typically have large parameter sizes and require training on large time series corpus, making efficiency critical. Notably, the paper you referenced (Shazeer et al., 2017) uses an entire section (Section 3) to discuss efficiency challenges in training MoE models, including considerations such as batch size, model parallelism, and network bandwidth. These discussions are engineering-focused—arguably even more so than our paper—but does that diminish their importance or exclude them from being considered technical contributions?
>
> Regarding the argument "the paper seems to be a relatively straightforward application of existing sparse MoE techniques". While it may sound easy to apply existing MoE techniques to TSFMs, making these techniques effective in practice is challenging. Our work represents a pioneering effort in successfully adapting sparse MoE to TSFMs. How do we define success in this context? An intuitive and primary criterion for success is the downstream model performance. For instance, the concurrent work Time-MoE, can hardly be considered successful, as its performance significantly trails behind Moirai-MoE. Notably, Time-MoE does not even surpass Moirai-Large, as demonstrated in both our zero-shot evaluations and the point forecasting results over 27 datasets reported by the Chronos Team (https://aws.amazon.com/blogs/machine-learning/fast-and-accurate-zero-shot-forecasting-with-chronos-bolt-and-autogluon/). In summary, while a straightforward application of existing MoE techniques to TSFMs may seem easy, achieving SOTA performance is technically non-trivial and depends critically on model design.
>
> Finally, as you acknowledged in the strengths part, we have put significant effort into the behavior of MoE models. This area of investigation has not been explored in the literature, including the Time-MoE work. Thus, this should be recognized as a meaningful contribution of our paper.
>
> **[C3] Cannot detect a significant difference between the proposed gating and the procedure in Shazeer et al, 2017. A more correct claim would be “we introduce a novel procedure for pre-training the gating function in a standard sparse mixture-of-experts model”.**
>
> Your suggested point is reasonable in certain respects; however, we would like to emphasize that the paper you mentioned (Shazeer et al., 2017) employs a linear projection as its gating, which relies on a randomly initialized, trainable weight matrix for expert assignment. In contrast, Moirai-MoE utilizes a gating function that is non-trainable and initialized based on clustering results from a pretrained model, not randomly. We believe this distinction marks a departure from the linear projection gating, effectively constituting a new gating function. We would appreciate further clarification regarding which aspects you consider identical between these two gating functions.
>
> **[C4] No confidence intervals and statistical tests.**
>
> Standard deviations and statistical significance are in Table 1: https://drive.google.com/file/d/1bwJ7dyji_OnSNkYXpA6IOpwnvF6nOZmS/view
>
> **[C5] Although the token clusters approach achieves the best performance, these results suggest that the training is not operating correctly.**
>
> Thank you for your comment. We do not believe our training is incorrect. Our routing mechanism is based on clustering results derived from a pretrained Moirai model's representations. If certain clusters exhibit similarity, the selection of experts would be naturally constrained, resulting in some experts being underutilized. Thus, the question here pertains to the representation similarity inherent in pretrained TSFMs. Existing research has investigated representation similarities within TSFMs. A recent study (https://arxiv.org/pdf/2409.12915v2) uses Centered Kernel Alignment to measure representation similarity, highlighting clear redundancy in TSFMs such as Chronos and Moirai. Additionally, another study (https://arxiv.org/pdf/2302.11939v2), specifically Section 7, reports that within-layer token similarity tends to increase in deeper Transformer layers. This finding aligns closely with our own observations in Moirai-MoE, where fewer experts are activated due to increased token similarities, particularly in deeper layers.

---

### Decision · Program_Chairs · 2025-05-01

**Decision:**

Accept (poster)

**Comment:**

This paper proposes MOIRAI-MOE, an extension of the MOIRAI time series foundation model that incorporates a sparse Mixture of Experts (MoE) architecture. The key motivation is to replace MOIRAI's predefined frequency-based grouping of time series with a data-driven, automated approach where different "experts" (specialized feed-forward networks) handle different types of token patterns. The authors introduce a novel, non-trainable expert gating mechanism based on clustering the attention outputs of a pre-trained dense MOIRAI model, arguing this is more robust than typical trainable linear gating layers sensitive to initialization. Extensive experiments on numerous datasets show MOIRAI-MOE variants achieving state-of-the-art performance compared to the original MOIRAI models and other foundation models like Chronos and TimesFM, often with fewer activated parameters during inference. Initial reviews acknowledged the paper's clarity, the importance of addressing data heterogeneity in TSFMs, and the extensive empirical evaluation. However, concerns were raised regarding the technical novelty (especially compared to prior MoE work like Time-MoE), the justification and novelty of the proposed cluster-based gating function, the lack of reported result variability, and observations of highly sparse expert utilization in deeper layers suggesting potential inefficiency or training issues.

In response, the authors provided detailed clarifications and significant additional experimental results. They defended the technical contribution by highlighting the non-trivial engineering effort required for successful MoE integration in TSFMs (contrasting with Time-MoE's lower performance) and emphasized the novelty of their non-trainable, cluster-initialized gating compared to standard trainable linear or hash-based approaches. They provided results with standard deviations and statistical significance tests, conducted new experiments comparing parameter-matched versions of MOIRAI-MOE and MOIRAI (favoring MOIRAI-MOE), presented results on pruning underutilized experts (showing performance stability), and offered explanations for the sparse expert activation based on representation similarity in deeper layers of pre-trained TSFMs, supported by existing literature. They also addressed minor issues regarding code access, visualizations, and experimental details. The rebuttal satisfied some reviewer concerns, particularly around variability reporting and code access. However, fundamental disagreements persisted regarding the novelty/justification of the gating mechanism versus prior art and the implications of the sparse expert utilization, with some reviewers remaining unconvinced by the provided explanations.